# Using smart speakers to contactlessly monitor heart rhythms

Anran Wang [1✉], Dan Nguyen[2], Arun R. Sridhar [2✉] & Shyamnath Gollakota [1✉]

Heart rhythm assessment is indispensable in diagnosis and management of many cardiac conditions and to study heart rate variability in healthy individuals. We present a proof-of-concept system for acquiring individual heart beats using smart speakers in a fully contact-free manner. Our algorithms transform the smart speaker into a short-range active sonar system and measure heart rate and inter-beat intervals (R-R intervals) for both regular and irregular rhythms. The smart speaker emits inaudible 18–22 kHz sound and receives echoes reflected from the human body that encode sub-mm displacements due to heart beats. We conducted a clinical study with both healthy participants and hospitalized cardiac patients with diverse structural and arrhythmic cardiac abnormalities including atrial fibrillation, flutter and congestive heart failure. Compared to electrocardiogram (ECG) data, our system computed R-R intervals for healthy participants with a median error of 28 ms over 12,280 heart beats and a correlation coefficient of 0.929. For hospitalized cardiac patients, the median error was 30 ms over 5639 heart beats with a correlation coefficient of 0.901. The increasing adoption of smart speakers in hospitals and homes may provide a means to realize the potential of our non-contact cardiac rhythm monitoring system for monitoring of contagious or quarantined patients, skin sensitive patients and in telemedicine settings.

[1] Paul G. Allen School of Computer Science and Engineering, University of Washington, Seattle, WA, USA. [2] Division of Cardiology, University of Washington, Seattle, WA, USA. ✉email: anranw@uw.edu; arun11@cardiology.washington.edu; gshyam@uw.edu

Clinical heart rhythm assessment depends on reliable acquisition of beat-to-beat intervals of the heart, also known as the R–R intervals. Physiologically, the R–R interval represents the time between successive ventricular depolarizations of the heart. Acquisition and assessment of R–R interval irregularity is necessary for diagnosing many cardiac arrhythmias and to study heart rate variability (HRV) in healthy individuals[1,2]. Although frequency domain analysis can estimate average heart rate in regular and quasi-periodic heart rhythm conditions, it fails when the rhythm is irregular, which is common in pathological conditions such as atrial fibrillation[3]. R–R intervals are conventionally measured by identifying individual heartbeats extracted using electrocardiography (ECG). This approach works for both regular and irregular rhythms but requires physical contact with the skin to operate.

A noncontact solution for heart rhythm monitoring offers several advantages. It can monitor infectious and contagious patients where cleaning of contact-based devices can be time consuming and burdensome[4,5], monitor patients in home isolation and quarantine settings, and benefit patients with skin allergies who are intolerant to wearable and contact-based devices[6]. Contactless rhythm acquisition may also be valuable in the modern telemedicine era, whereby patients' self-administered rhythm analysis are communicated to their physician. The benefits of a self-administered test are numerous, and may include the ability to connect patients living in rural areas to physicians, screening patients for atrial fibrillation remotely, and obtaining clinical trial data without the need for an in-person visit.

The widespread adoption of high-quality smart speakers equipped with multiple microphones presents a unique opportunity for contactless monitoring of human body and internal organ functions. Google Nest smart devices can already determine a user's distance on its smart speaker by emitting soft, inaudible acoustic signals and analyzing their reflections from the human body[7,8]. Apple HomePod and Amazon Echo devices support an array of six and seven microphones, respectively, that are used for sophisticated acoustic processing[9].

Here, we describe a proof-of-concept contactless system for monitoring cardiac rhythm using smart speakers that can identify individual heartbeats in both regular and irregular rhythms. Our algorithms extract both heart rate and R–R intervals by transforming a smart speaker into a short-range active sonar system. An active sonar-based approach to contactless monitoring has the distinct benefit of scalability vis-a-vis smart speakers. Unlike Doppler radar[10–13] and optical vibrocardiography[14–16], active sonar hardware components (i.e., multiple microphones and speaker) are ubiquitous in smart speakers. Further, in contrast to approaches that use facial photoplethysmographic signals[17,18], which raise privacy issues due to their use of cameras, active sonar can operate using inaudible acoustic signals and does not require the capturing of audible sounds.

At a high level, a smart speaker emits 18–22 kHz inaudible sound signals that are reflected off the human body and received by a microphone array. We designed algorithms to (1) analyze these signals and detect the subtle motion of the chest wall caused by the heart's apical impulse as well as by arterial pulsations on the body's surface, and (2) separate these signals from much larger breathing motions and ambient noise. We show that a smart speaker running our algorithms that is placed in front of a subject less than a meter away can identify individual heartbeats and extract heart rate and R–R intervals for both healthy participants and patients with different cardiac abnormalities. These data could be used for studying heart rhythms, detecting cardiac arrhythmias, and determining HRV.

## Results

**Concept and algorithms.** Prior work has focused on contactless monitoring of breathing signals using active sonar on smart devices[19–22]. Recent work[23] computes heart rate using smart phones from 5 to 30 cm, but assumes that the heartbeats are regular and thus uses frequency domain analysis to extract the heart motion from the fundamental frequency and its harmonic components. This approach however does not work with irregular heart rhythm since there is no well-defined peak in the frequency domain and the energy is spread across a range of frequencies. Extracting irregular beats is difficult using acoustic signals since heartbeats result in a 0.3–0.8 mm motion on the surface of the human body[24]; this is an order of magnitude smaller than the wavelength of sound at our operational frequencies. Further, commodity smart speakers are designed primarily to transmit in the audible frequencies, and the inaudible frequencies they support have a limited bandwidth—4 kHz bandwidth across 18–22 kHz—with a nonideal frequency response. Unlike ultrasonic devices[25], commodity smart devices also have a limited sampling rate, about 48 kHz, that produces a low signal-to-noise ratio, making it difficult to achieve the high temporal resolution required to measure the precise timing of each heartbeat. Another complicating factor is that breathing creates a much larger motion than heartbeats on the surface of the body. Though respiration rates are typically lower than heart rates, respiration is not a perfect sinusoidal motion since inhalation and exhalation durations can differ (Fig. 1A). This creates high-frequency components in the breathing motion that interfere with the minute heartbeat motion. At low signal-to-noise ratios, this prevents the latter from being reliably separated in the frequency domain using filtering (Fig. 1B); when the heart signal is weak and overwhelmed by interference from breathing motion, it becomes challenging to extract individual heartbeats in irregular rhythm.

Our smart speaker-based sonar system generates frequency modulated (FM) continuous wave (FMCW) signals, with the frequency linearly increasing from 18 to 22 kHz. We extract individual heartbeats from reflections of these transmissions captured by a microphone array. We first preprocess the received signal at each microphone to filter out the audible frequencies to remove background noise. We then extract the impulse response of the acoustic channel which represents the times of arrival of the various reflections from the speaker to the microphone. Since cardiac motion is minute, it can be drowned out by reflections corresponding to coarse motion from distant locations. Therefore, we perform echo suppression to eliminate echoes arriving from distances greater than 1 m (Fig. 1C).

We then separate the heart rhythm from breathing motion. Heart rhythm can be irregular, and breathing motion is not a perfect sinusoidal signal. Therefore, filtering alone is not effective. We introduce an adaptive learning-based beamforming algorithm that maximizes the signal-to-interference and noise ratio (SINR) by aligning heartbeat signals across microphones and frequencies while minimizing the interference from breathing motion and noise. The adaptive beamformer uses complex weights to combine the signals from different microphones across frequencies. To compute the weights, we formulate an optimization function that we solve using a gradient ascent algorithm[26]. Since we do not assume a priori periodic structure to the heart rhythm, the learning algorithm can erroneously detect high frequency, impulse-like signals caused by abrupt breaths or interference in the environment. We introduce regularization parameters in the optimization function by penalizing such abrupt changes (see "Methods").

Finally, we segment the resulting heart rhythm signal into individual heartbeats. Since beamforming can be imperfect

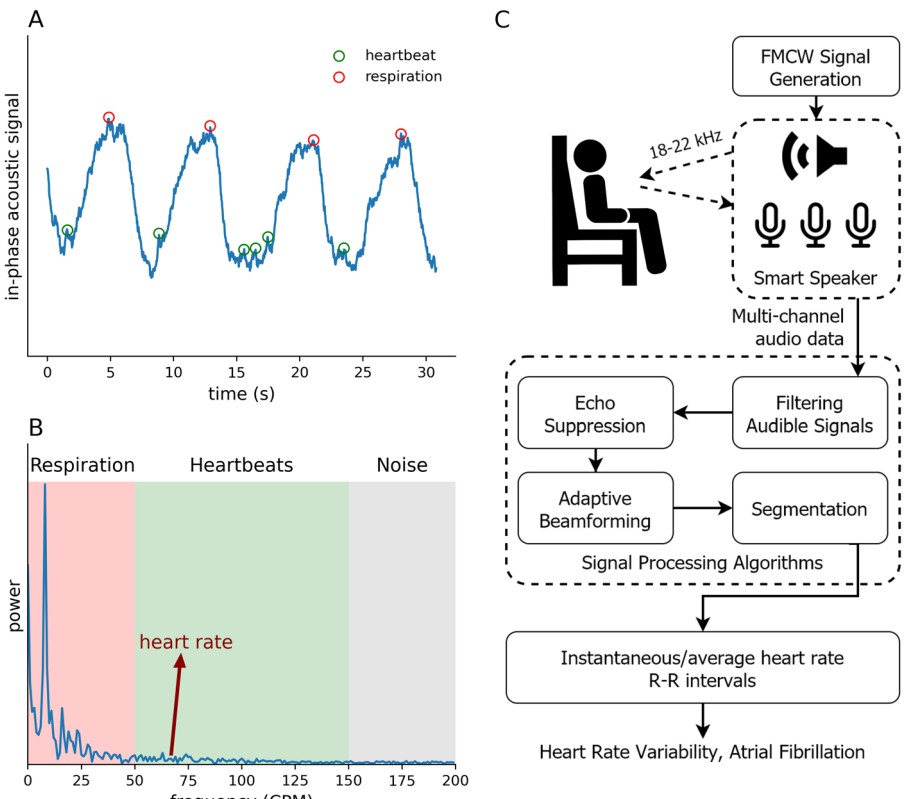

**Fig. 1 The processing pipeline of our system is able to extract the tiny motion of heartbeats from the raw active sonar signal. A** The displacement from respiration and heartbeat in the in-phase component of the raw active sonar signal. While breathing motion (blue curve) is strong, the heartbeats (red circle) are weak and not reliably observable in this signal. **B** The frequency domain with the respiratory frequency range, heart rate range, and high-frequency noise. The respiratory signal strength is much higher and its harmonics spread into heart rate frequencies preventing the latter from being reliably extracted by just filtering. **C** The different algorithms in our system to separate heart motion from respiration and extract individual heartbeats.

(see Supplementary Fig. 1), we still confront the challenge of nonnegligible residual interference from respiration motion, which shifts the heart signal back and forth between the in-phase and quadrature-phase components of the acoustic signal (see Fig. 2). Our algorithm simultaneously identifies the segmenting points and the shift in each segment. We do this by (1) comparing adjacent segments to account for different segment lengths due to irregular R–R intervals and (2) tracking the shift between in-phase and quadrature-phase components caused by residual breathing motion. Once we identify each beat segment, we compute the heart rate and R–R intervals.

**Testing with healthy participants**. We recruited a cohort of 26 voluntary participants who had no prior history of cardiac conditions. The median age of the participants was 31 [interquartile range (IQR), 8.5] years and body mass index (BMI) was 22 (IQR, 3). The female-to-male ratio was 0.6 (see Supplementary Table 1).

Participants were fitted with a Polar H10 Sensor System (Polar Electro, Kempele, Finland) that measures ECG and outputs the heart rate and R–R intervals. We used the ECG sensor to gather ground truth data for the study. All testing was performed in a private room at the University of Washington, where participants sat upright on a chair by a table on which our prototype smart speaker was placed. The testing was conducted with the clothing the participants were already wearing indoors such as blouses, tops, T-shirts, and button downs made with different fabric materials. Participants took a series of 1-min measurement sessions, where they were asked to sit still and breath normally. For each healthy participant, we conducted a total of seven 60-s

sessions. In the first three, the smart speaker was placed in front of the participant's chest at the nipple level, at a distance of 40, 50, and 60 cm. For the fourth session, the smart speaker was pointed 10 cm above the participant's chest at a distance of 50 cm. For the fifth, the smart speaker was pointed toward the chest but at an angle of 20° and a distance of 50 cm. In the sixth, measurements were conducted at a distance of 50 cm, while jazz music played at around 75 dB(A) sound power level from a distance of 5 m. In the final session, participants were asked to jog in place to increase their heart rate above 110 beats per minute (BPM) before starting measurements at a distance of 50 cm.

We computed the average heart rate by counting the number of heartbeats over a period of 60 s and compared it to the heart rate output by the ECG device. Figure 3A shows the scatter plot of the heart rates across all participants and sessions. Measurements from the smart speaker and the ECG sensor had intraclass and concordance correlation coefficients (CCCs) of both 0.983. Figure 3B shows the cumulative distribution function of the error in the heart rate. The median absolute error (MAE) was 1 BPM, with a 90th percentile error of less than 4 BPM. We also compared the R–R intervals output by the smart speaker and the ECG sensor. The intraclass correlation (ICC) coefficient and CCC between the two measurements were 0.929 and 0.927, respectively (Fig. 3C). The MAE in the R–R intervals was 28 ms, with a standard deviation of 49 ms, and the 90th percentile error was 75 ms (Fig. 3D). The mean absolute error in the R–R intervals as a percentage of the ground truth R–R interval was 3.6% with a standard deviation of 4.3%.

As the distance from the speaker to the participant increased, the acoustic signal attenuated, increasing errors. As Fig. 4A

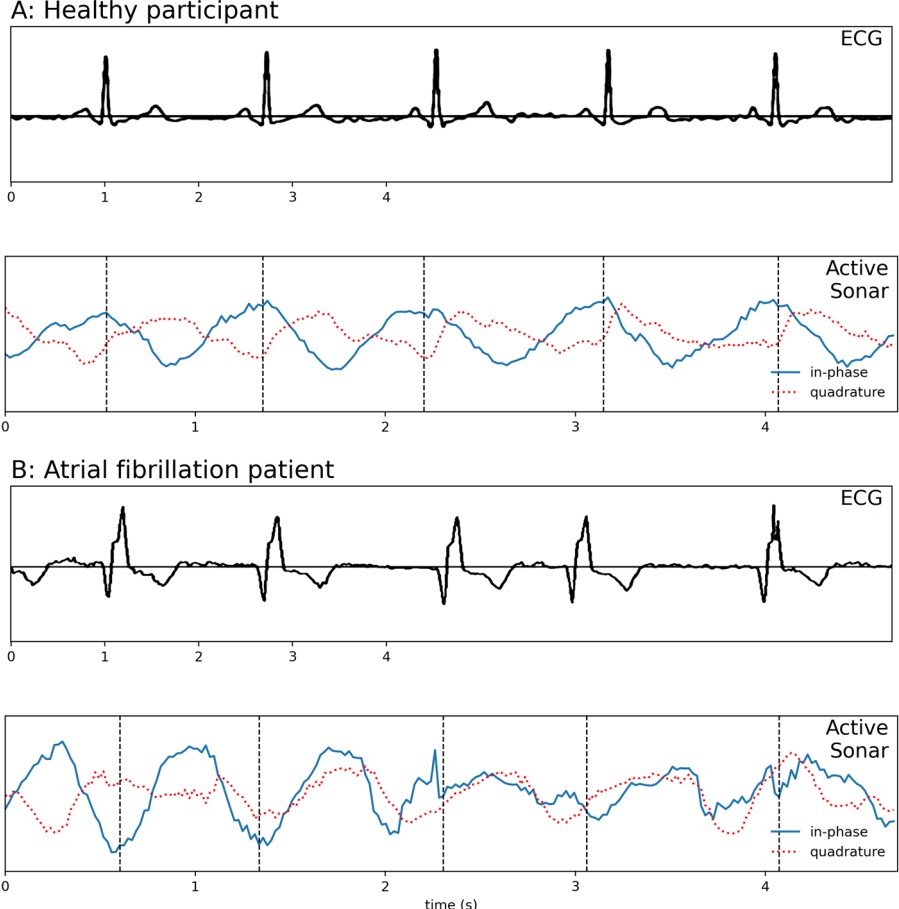

**Fig. 2 Example heart rhythm waveforms extracted by our system along the ground truth ECG waveforms.** The acoustic waveforms include both in-phase and quadrature-phase components after beamforming. The cardiac rhythm signal shifts between the in-phase and quadrature-phase components due to residual respiration motion that remains after beamforming. The vertical dotted lines show the segments computed by the segmentation algorithm, which combines data from both in-phase and quadrature-phase components. The figure shows the waveforms for a **A** healthy participant and **B** atrial fibrillation patient.

shows, when the distance increased from 40 to 60 cm, the median error in the R–R intervals increased from 25 to 33 ms. The median error was 26 ms when the speaker pointed 10 cm above the chest level (Fig. 4B) and 31 ms when the speaker pointed at an angle of 20° from the chest (Fig. 4C). This demonstrates that our adaptive beamforming algorithm provided some tolerance to imperfect alignments of the smart speaker system. The algorithm is also resilient to larger angles with the smart speaker placed to the left and right of the participant; the error however is high when placed behind the participant, facing their back (Supplementary Fig. 2).

Figure 4D shows that background music increased the median error from 25 to 32 ms; this is likely due to residual high-frequency components and nonlinearity of the phone emitting the music. Since breathing is more pronounced after exercise, it can create larger amounts of interference; the median error was 32 ms after exercising, in contrast to an error of 25 ms during the rest state (Fig. 4E). Finally, R–R intervals for female participants showed a median error of 30 ms versus 27 ms for male participants (Fig. 4F). The error also slightly increases with BMI (Supplementary Fig. 3).

**Testing with cardiac patients**. We also tested system performance for hospitalized cardiac patients ($n = 24$). Once enrolled in the study, the patients' existing telemetries were reviewed by a medical doctor (D.N.), and the patients were adjudicated into

either a regular rhythm category (sinus rhythm, atrial flutter with regular conduction, ventricular paced, or atrioventricular paced) or an irregular rhythm category (atrial fibrillation or atrial flutter with variable conduction). Table 1 shows baseline demographic and clinical data for cardiac patients stratified by heart rhythm. Patients in the irregular rhythm cohort were more likely to have a history of atrial fibrillation and more likely to be female. Age, BMI, reason for hospitalization, medical comorbidities, and cardiac medications were uniform between the regular and irregular rhythm cohorts. Since prior audiocardiography work showed poor results in extreme obese patients[27], we excluded patients whose BMI exceeded 35 for this study but evaluated them in a separate study described later.

To obtain ground truth heart rate and R–R interval data for comparison, half the patients were fitted with a chest-worn Polar H10 Sensor System (Polar Electro, Kempele, Finland). Patients unable to wear the chest band due to discomfort, recent thoracic surgery, or poor ECG signal acquisition ($n = 12$) were fitted with a fingertip-worn CorSense monitor (Elite HRV, Asheville, NC, USA). These data were downloaded in real time to a bluetooth-connected smartphone using the HRV+ mobile app (Elite HRV, Asheville, NC, USA). The rationale behind this method is that hospital telemetry software does not allow for digitalization and storage of the R–R interval data. Previous studies have demonstrated portable HRV devices to have acceptable error compared to gold standard ECG monitoring[28].

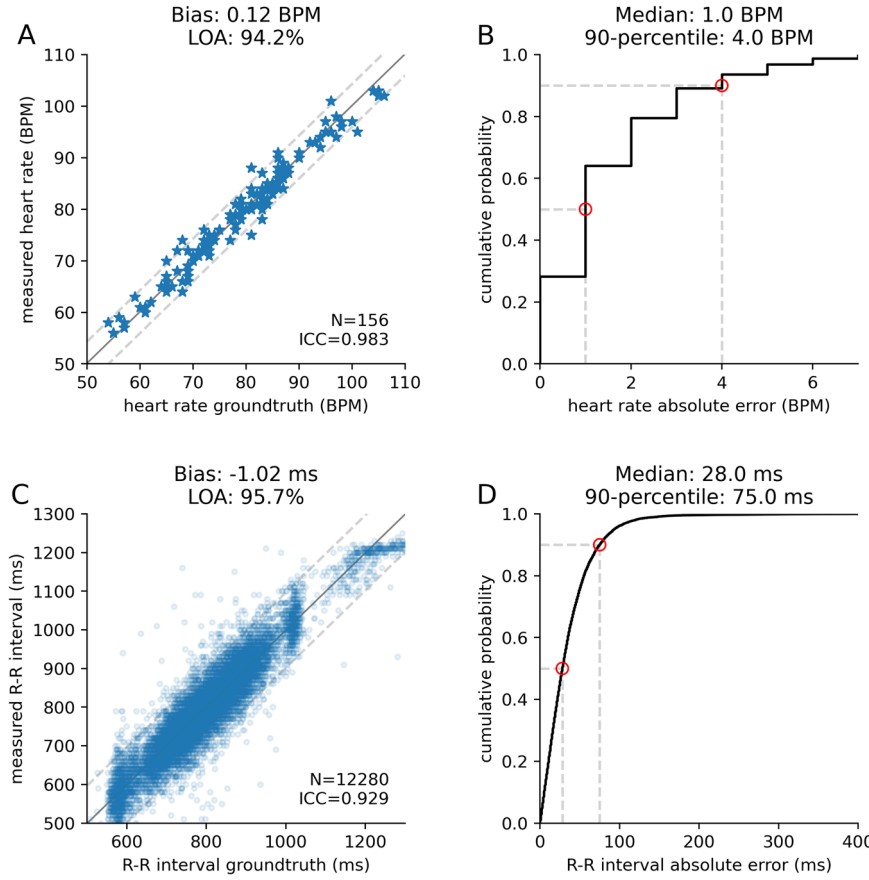

**Fig. 3 Evaluation of heart rate and R–R interval accuracies among the healthy participants. A** Scatter plot of average heart rate in beats per minute (BPM) compared with ground truth. **B** Cumulative distribution function (CDF) of the absolute heart rate error. **C** Scatter plot of R–R intervals compared with ground truth. **D** The CDF of the absolute R–R interval error. Limits of agreement (LOA) were computed using twice the standard deviation (gray dashed line in **A** and **C**. Median and 90th percentile values are noted in red circles in **B** and **D**.

Patients were positioned sitting vertically on the hospital beds in their own room and the smart speaker system was placed around 50–60 cm from them, with the speaker inlet pointed at the chest at the level of the nipple. Ambient noise sources (e.g., television) were turned off and family members and visitors of patients who were required to stay in the room were asked to sit at least 2 m away from the smart speaker during the sessions. Data were acquired from the smart speaker system in five sessions, each lasting 60 s. During each session, patients were instructed to remain still. All patients tolerated the data acquisition process; however, data acquisition was prematurely terminated for one patient due to developing nausea related to a prior medical condition.

Figure 5A, B shows system performance in computing the average heart rate across all cardiac patients. The MAE in the heart rate was 2 BPM, with a 90th percentile error of less than 3 BPM. For R–R intervals, the ICC and CCC were 0.901 and 0.898, respectively (Fig. 5C). The MAE in the R–R intervals was around 30 ms, with a standard deviation of 67.2 ms, and the 90th percentile error was less than 93 ms (Fig. 5D). The mean absolute error in the R–R intervals as a percentage of the ground truth R–R interval was 4.0% with a standard deviation of 7.6%.

Focusing on irregular heartbeats, the mean absolute R–R interval error among patients with atrial fibrillation instances was 35 ms with ICC and CCCs of 0.891 and 0.890, respectively. Higher median R–R intervals correspond to higher 90th percentile error (Supplementary Fig. 4). There was no noticeable decrease in accuracy among those with irregular rhythms compared to those with regular rhythms. Within the context of

clinical practice, it is unlikely that this magnitude of error would result in diagnostic errors for detecting atrial fibrillation where R–R interval variation less than 50 ms is often not clinically important. In atrial fibrillation, the R–R interval widely varies from beat to beat and standard deviations range between 95 and 233 ms in different physiological states[29]. Proper diagnosis of rhythm disorders relies on the ability to detect temporally disparate R–R intervals, rather than precise R–R interval measurement.

The time series plots in Fig. 6A–E show the R–R intervals for atrial fibrillation instances. Both ground truth and smart speaker data showed noticeable variation in R–R intervals, which is indicative of irregular heartbeats. Figure 6F shows an instance of respiratory sinus arrhythmia where both data streams showed that the R–R interval duration decreased with inspiration and increased with expiration. Figure 6G corresponds to a patient with an implanted permanent cardiac pacemaker and a paced rhythm. The patient in Fig. 6H had an intrinsic rhythm (nonpaced rhythm) and this patient had mild variations in the R–R intervals with a standard deviation less than 10 ms. This low level of HRV is not uncommon. We collected data from patients in the cardiac floor of our tertiary care medical center with a variety of cardiac conditions, which included cardiac conduction disorders, arrhythmias, cardiomyopathy, as well as valvular disorders. Many of these cardiac conditions directly or indirectly affect the HRV. Respiratory sinus arrhythmia, which is a major cause of HRV, becomes less common with age[30] and is less prevalent in patients with diabetes due to autonomic neuropathy[31]. Our hospitalized population had a mean age of

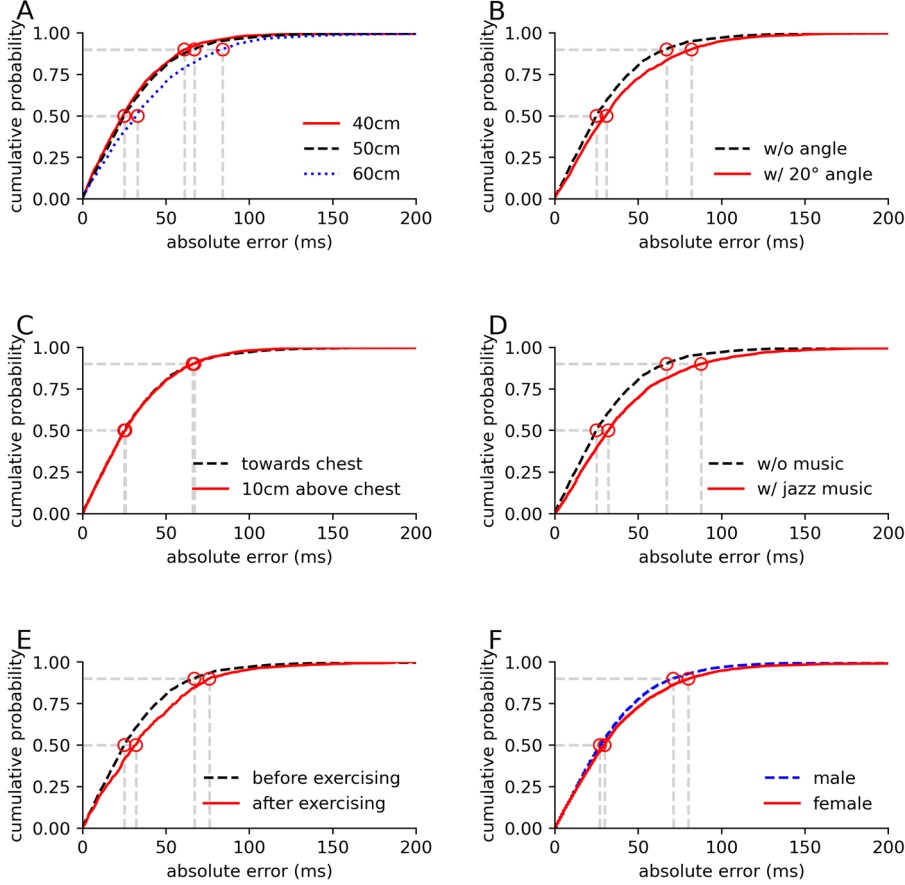

**Fig. 4 Cumulative distribution functions (CDFs) of absolute error in R–R intervals in different sessions with healthy participants. A** CDFs at different distances (red line, black dashed line, and blue dotted line represent 40, 50, and 60 cm, respectively) from the speaker, **B** CDFs with the speaker pointing left at a 20° angle (red line) from the chest, compared to right in front of the chest (black dashed line), **C** CDFs with the speaker pointing 10 cm above chest (red line), **D** CDFs with music playing in the background (red line), **E** CDFs after exercising in the last session (red line), and **F** CDFs across genders (red line and dashed blue line represent female and male, respectively).

63.2 years in the regular rhythm group and 68.0 in the irregular rhythm group, and there were a total of 5 out of 24 patients with diabetes. In addition, medications that influence vagal tone, such as beta blockers, digoxin, and opiate pain medications, may decrease sinus arrhythmia[32]. Our sample of hospitalized cardiac patients often had multiple factors which could reduce HRV (Fig. 6G, H).

**Effect of extreme obesity.** The above study excluded cardiac patients with a BMI greater than 35. Next, we evaluated the algorithm's performance for five extreme obese hospitalized cardiac patients with BMIs between 36 and 40.4 (median BMI of 38.6). Our algorithm could extract cardiac rhythm signals for only one of these five participants, likely because excessive adipose tissue dampens motion of the heart at the body's surface. This effect has also been shown in the past to limit the use of audio-cardiography[27] and optical vibrocardiography[33] for cardiovascular examination of the severe obese. These findings are in line with our results with healthy participants, where the error was slightly higher for female participants (Fig. 4F).

## Discussion

Smart speaker technology is rapidly evolving and may provide a reliant and convenient platform for the next generation of health monitoring solutions[22,34]. Indeed, the increasing adoption of smart speakers in hospitals[35] and homes[36] could provide a means

to realize the potential for our contactless cardiac rhythm monitoring system.

The ability to monitor cardiac rhythm using smart speakers raises privacy concerns. The short-range nature of active sonar, however, can protect privacy since it requires the direct engagement and implicit consent of the user, who must be within a meter of the speaker and stay still. The 18–22 kHz acoustic frequencies we use in our system also contain little information about audible sounds in the environment. Finally, smart speaker manufacturers do not give third-party app developers access to raw acoustic signals from individual microphones. Consequently, the smart speaker manufacturers can implement and deploy this capability in a manner that balances the needs and concerns of patients, healthcare providers, and privacy advocates.

Certain differences between healthy participants and cardiac patients may impact the fidelity of heart rate and R–R interval acquisition using smart speakers in patients with cardiovascular disease. Patient factors that alter arterial vessel and ventricular compliance and medical treatments that alter thoracic anatomy and ventricular contractility are more prevalent in hospitalized cardiac patients. For instance, increased age and hypertension both cause blood vessel stiffening via vessel fibrosis, collagen deposition, and elastin degradation within the vessel wall[37], which subsequently reduce pulse wave velocity and radial vessel motion. Patients with hypertension or coronary artery disease may develop increased ventricular stiffness in a process known as diastolic dysfunction[38]. Our cohort had several patients with

**Table 1 Demographic information for hospitalized cardiac patients.**

|  | Regular rhythm (n = 18) | Irregular rhythm (n = 6) |
|---|---|---|
| Baseline characteristics, mean ± SD |  |  |
| Age (years) | 63.2 ± 13.4 | 68.0 ± 7.6 |
| Height (cm) | 172.5 ± 8.0 | 174.2 ± 14.0 |
| Weight (kg) | 82.0 ± 17.6 | 74.0 ± 18.3 |
| BMI (kg/m$^2$) | 27.5 ± 5.0 | 24.3 ± 4.7 |
| Female (n, %) | 2 (11.1%) | 2 (43.3%) |
| Reason for admission, n (%) |  |  |
| Acute coronary syndrome | 4 (22.2) | 0 (0.0) |
| Heart failure exacerbation | 5 (27.8) | 4 (66.7) |
| Cardiogenic shock | 4 (22.2) | 1 (16.7) |
| Valve disease | 1 (5.6) | 1 (16.7) |
| Other | 4 (22.2) | 0 (0.0) |
| Comorbidities, n (%) |  |  |
| Hypertension | 8 (44.4) | 3 (50.0) |
| Hyperlipidemia | 6 (33.3) | 2 (33.3) |
| Atrial fibrillation | 6 (33.3)[a] | 6 (100.0) |
| Atrial flutter | 1 (5.6) | 0 (0.0) |
| Conduction system disease | 3 (16.7) | 1 (16.7) |
| Coronary artery disease | 6 (33.3) | 1 (16.7) |
| Diabetes mellitus | 4 (22.2) | 1 (16.7) |
| Congestive heart failure | 14 (77.8) | 5 (83.3) |
| Valvular disease | 7 (38.9) | 4 (66.7) |
| Heart transplant | 2 (11.1%) | 0 (0.0%) |
| Stroke/transient ischemic attack | 4 (22.2) | 2 (33.3) |
| Obstructive sleep apnea | 3 (16.7) | 2 (33.3) |
| Chronic kidney disease | 6 (33.3) | 1 (16.7) |
| Smoker |  |  |
| Current | 2 (11.1) | 0 (0.0) |
| Former | 3 (16.7) | 3 (50.0) |
| Medications, n (%) |  |  |
| ACE inhibitor | 4 (22.2) | 1 (16.7) |
| Angiotensin receptor blocker | 3 (16.7) | 2 (33.3) |
| Aldosterone antagonist | 5 (27.8) | 2 (33.3) |
| Loop diuretic | 8 (44.4) | 3 (50.0) |
| Beta blocker | 10 (56.6) | 2 (11.1) |
| Calcium channel blocker | 1 (11.8) | 2 (28.6) |
| Antiarrhythmic drug | 0 (0.0) | 0 (0.0) |
| Statin | 12 (66.7) | 2 (33.3) |
| Digoxin | 3 (16.7) | 0 (0.0) |
| Oral anticoagulant | 7 (38.9) | 5 (83.3) |
| Aspirin | 9 (50.0) | 3 (50.0) |

[a]These are atrial fibrillation patients, but at the time of data acquisition they were noted to be in regularized rhythm.

advanced congestive heart failure and reduced ventricular function; these patients may have displaced and diminished apical impulses due to left ventricular dilation[39] and are often on medications that further reduces cardiac contractility, such as beta blockers and antiarrhythmic drugs. Lastly, patients recovering from cardiogenic shock or advanced heart failure who received a heart transplant as part of their treatment may have distorted thoracic anatomy due to acute or chronic postsurgical inflammatory changes. These anatomical and physiologic differences may explain the performance variation of our smart speaker system between healthy patients and patients with cardiovascular diseases.

Our study has the following limitations. Our beamformer algorithm assumes that the average heart rate falls in the 60–150 BPM range. This is not a hard threshold and our bandpass filter can detect cardiac signals between 50 and 60 BPM (Fig. 3A). If the heart rate is much lower, it may however not detect any cardiac signal or may amplify spurious noise. Like Doppler radar[10] and optical vibrocardiography[17], our system requires participants to remain still for the duration of the examination and assumes that the measuring device neither moves nor is prone to vibrations. Movements can affect the ability to extract the cardiac rhythm (see Supplementary Fig. 5). Performance results with the healthy cohort showed the system's reliability across diverse participant clothing, which included a single layer of shirts and tops that were not tightly fit, and many of the hospitalized patients wore loose gowns. While loose clothes can affect accuracy, the degradation was not drastic: the median and 90th percentile absolute R–R interval errors changed from 24 to 26 ms and 84 to 80 ms, respectively, for two participants who participated with both tight and loose clothes. However, multiple layers of clothing can limit the ability to extract heart motion since sound attenuates through thick fabric. Since we eliminate echoes at distances greater than 1 m, family members of the hospitalized cardiac patients could be in the same room during the study. At this time, our system is designed for spot monitoring of a single participant. Further hardware and software enhancements could enable continuous monitoring. To improve signal strength and range, the smart speaker hardware may need directional tweeter which can rotate to the direction of interest as well as speakers with a better response at the target frequencies and microphones with higher sampling rates and bit resolutions. New smart speaker models have rotatable directional tweeter capabilities (e.g., Amazon Echo Show 10) and have microphones and speakers that are designed to operate at the target frequencies; in contrast, our hardware has a 10–15 dB degradation at 18–22 kHz. Multiple participants could be supported using FMCW algorithms that use breathing motion to track the location of each participant and then separate cardiac signals from different distances[20].

Our smart speaker prototype has a sampling rate of 48 kHz and uses 18–22 kHz acoustic transmissions which are generally inaudible to adults but can be audible to the younger population. Commercial smart speakers like Google Nest support acoustic frequencies between 25 and 30 kHz, which are inaudible across the age spectrum and could be used to enable cardiac rhythm monitoring using our algorithms. Frequencies higher than 30 kHz require specialized hardware and also limit the range of the system. The World Health Organization recommends a noise limit of 85 dB(A) over an average duration of 8 working hours[40]. Our exposure intensity was 75 dB, (around 66 dB(A) at 20 kHz and 50 cm), which is much less than that. Short-time exposure to high frequency also does not affect the hearing capability of infants[41]. Pets have even higher sensitivity to ultrasound as high as 64 kHz[42] and sound around 40 kHz can potentially interrupt their sleep[43] and cause feline audiogenic reflex seizures for cats[44]. However, sounds in the 18–30 kHz are not known to affect animals. Prior active sonar studies report that 18–22 kHz FMCW signals did not elicit reaction from dogs[19].

Radar-based systems use radio signals with large bandwidth and use custom hardware that is not pervasive in smart speakers. Prior radar-based studies report a median R–R interval error of 8–44 ms for healthy participants[13,45–47] and 186 ms for cardiac patients with atrial fibrillation[10]. Our sound-based system instead uses active sonar algorithms, hardware that is pervasive in smart speakers, and is designed to achieve low errors for both regular and irregular rhythm. Finally, ECG captures the electrical activity in the heart that includes information about the P wave, QRS complex, and T wave. Our system is limited to providing the heart rate and R–R intervals. The R–R intervals can also be identified visually using a single-lead ECG signal. In contrast, the cardiac motion appears in both the in-phase and quadrature components

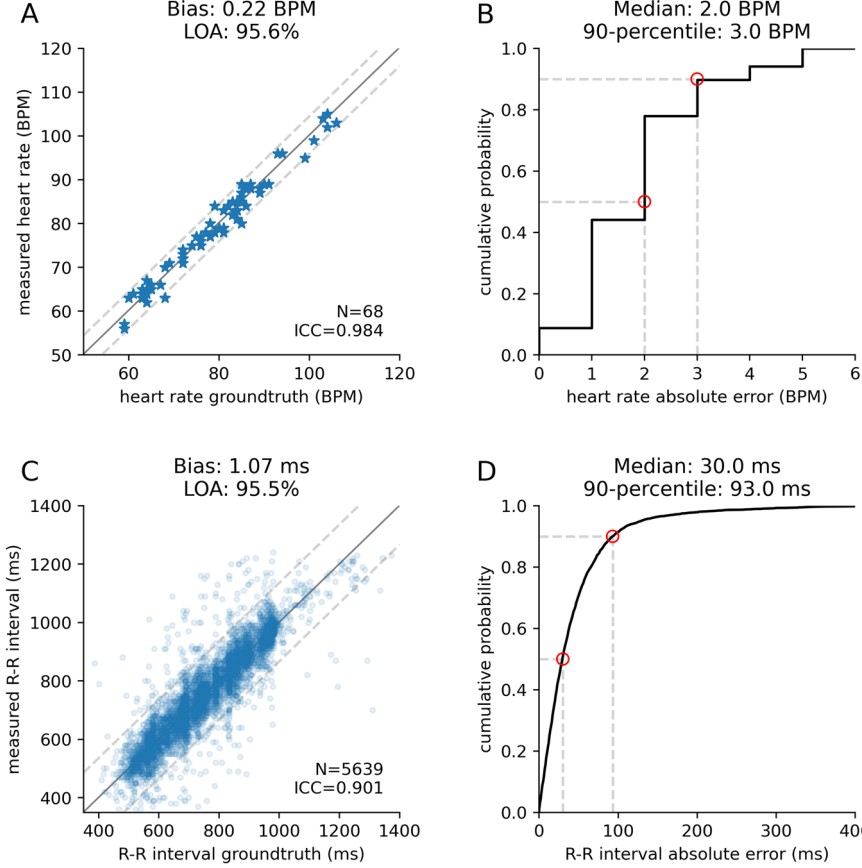

**Fig. 5 Evaluation of heart rate and R–R interval accuracies among the hospitalized cardiac patients. A** Scatter plot of average heart rate in beats per minute (BPM) compared with ground truth. **B** The cumulative distribution function (CDF) of the absolute heart rate error. **C** Scatter plot of R–R intervals compared with ground truth. **D** The CDF of the absolute R–R interval error. Limits of agreement (LOA) were computed using twice the standard deviation (gray dashed line in **A** and **C**). Median and 90th percentile values are noted in red circles in **B** and **D**.

of the active sonar signal and requires computationally combining both these components to compute R–R intervals.

We build on prior work that uses ultrasonic devices[25,48]. These systems use custom hardware with ultrasound frequencies and sampling rates not supported by commodity smart speakers, transmit signals at a sound pressure level of 105 dBm at 30 cm[48], which is about 300 times higher than that used by our prototype, achieve a limited range of 10–20 cm, and have not been clinically evaluated. Our system addresses these limitations and shows the feasibility of noncontact monitoring of individual heartbeats in both healthy and cardiac patients using smart speakers.

In summary, we presented a proof-of-concept system that can extract cardiac rhythm data using smart speakers. The ability to compute R–R intervals and HRV has proven to be clinically useful in distinguishing between atrial fibrillation and sinus rhythm[49]. It has also been used to monitor stress, anxiety, and the general health of the autonomic nervous system[1]. Further studies are required to determine the technology's utility for these and other potential scenarios.

## Methods

**Study design**. Cardiac patients were enrolled prospectively from the acute care general cardiology unit at the University of Washington Medical Center, a tertiary academic medical center in an urban area. All patients' heart rates and rhythms were continuously monitored in this unit using hospital-commissioned, three-lead surface electrode telemetric monitoring systems.

Patients were eligible for inclusion if they were older than 18 years of age and able to provide informed consent. They were excluded if they were unable to sit still for more than 15 min, demonstrated cardiopulmonary instability, or had altered mental status as determined by a medical doctor (D.N.). Randomization was not applicable, and study investigators were not blinded. Once enrolled in the study,

patients had their clinical variables—age, gender, height, weight, BMI, medications, and medical comorbidities—abstracted from their electronic medical records. This study was approved by the University of Washington Institutional Review Board, and all relevant ethical regulations were followed and informed consent was obtained.

In the study, we use the Elite HRV CorSense PPG and Polar H10 ECG sensors for ground truth. PPG sensors are known to produce comparable R–R interval accuracies to ECG, with high correlation coefficients between 0.968 and 0.998[50,51]. To verify this, we performed a comparison test between the ground truth sensors on two healthy participants and noted that the mean absolute R–R interval difference was 11 ms.

**Smart speaker prototype**. Though smart speaker companies have access to individual microphone data from the microphone array, these data are not currently provided to third-party developers to protect user privacy. Therefore, we prototyped our system using an off-the-shelf, seven-microphone array, which had an identical microphone layout and sensitivity to the Amazon Echo Dot[9] but can output raw recorded signals. The prototype consisted of a commercial UMA-8-SP USB circular array with seven microphones with a 4.3 cm separation, similar to an Amazon Echo Dot; a PUI Audio AS05308AS-R speaker; and a 3D-printed case that held the microphone array and the speaker next to each other (see Supplementary Fig. 6). The smart speaker was connected to a computer via USB as an external sound card device, where we played and recorded sounds at a sampling rate of 48 kHz and a sound pressure level of around 75 dB at a distance of 50 cm. A similar setup and hardware were used in smart speaker research due to the constraints imposed by smart speaker companies[22,52,53].

The minimum distance resolution achieved by our system depends on various factors that affect phase error: hardware components, circuit design and interference control, operating system and driver to support high-throughput audio signals, and the algorithm itself. The mean phase error on our acoustic hardware is ~0.05 radian in an empty room. Assuming signals from each of the seven microphones are independent, the corresponding mean displacement error, with ideal beamforming, is around 0.025 mm. Note that this is an ideal distance resolution for our specific hardware and is likely better for consumer smart speakers with better hardware.

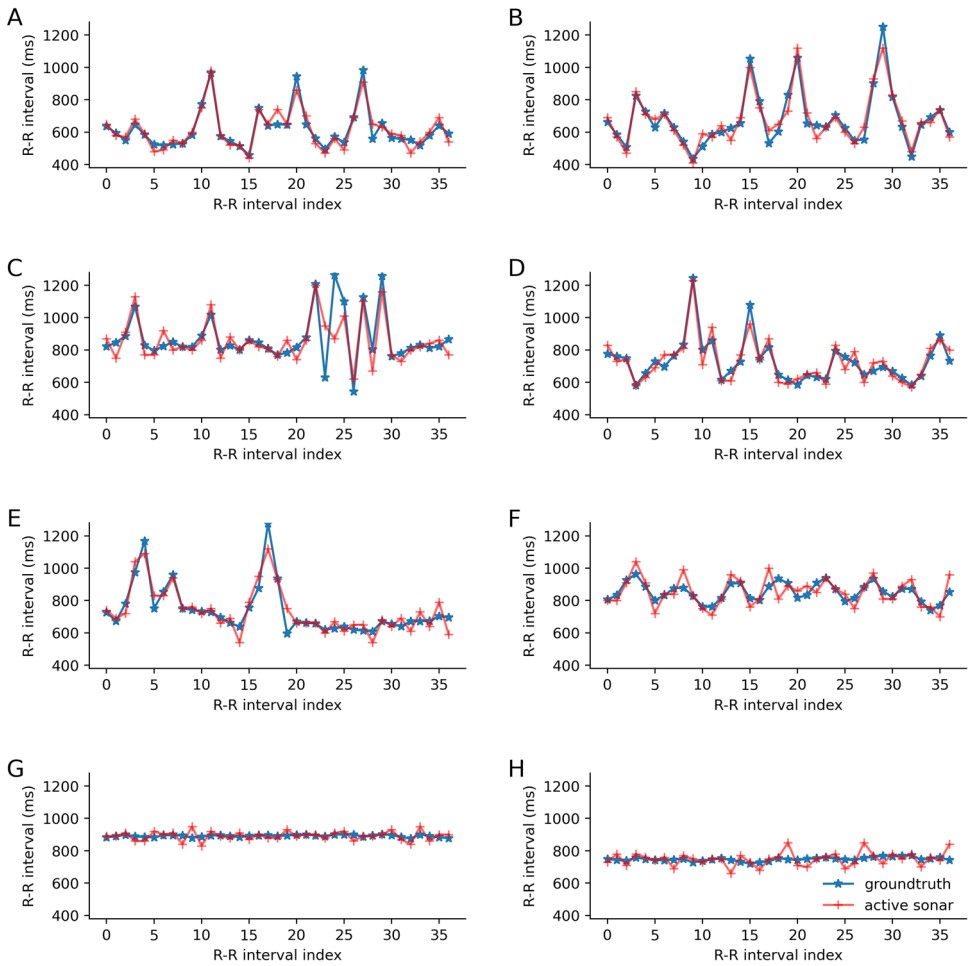

**Fig. 6 Example plots showing the time series of R–R intervals for cardiac patients, where red and blue lines represent our active sonar system and ground truth, respectively. A–E** five atrial fibrillation patients, **F** a patient with respiratory arrhythmia, and **G**, **H** two patients with sinus rhythm without arrhythmia.

**Extracting cardiac rhythm using active sonar**. We generated a linear FMCW chirp block with a duration of $T = 50$ ms, between $f_0 = 18$ kHz and $f_0 + F = 22$ kHz, and played it in a loop through the speaker. While we did not perform the traditional FMCW processing and other signals including white noise could be used[22], we used FMCW signals since they provide good spectral efficiency. Mathematically, an FMCW signal is given by

$$x(t) = \cos(2\pi f_0 t + \pi \frac{F}{T} t^2), t \in [0, T) \qquad (1)$$

We performed a discrete Fourier transform (DFT) on this signal to extract its frequency domain representation. We then computed the phase of the transmitted FMCW signal in the frequency domain within $[f_0, f_0 + F]$ as $\phi_{\text{FMCW}}(f)$, which we next used in our preprocessing algorithm.

*Preprocessing and echo suppression*. We first preprocessed the received signal at each microphone to extract the impulse response of the acoustic channel. We then suppressed the echoes that arrived from large distances.

To compute the impulse response of the acoustic channel on each microphone, we performed DFTs over signal blocks of duration $T$ with a sliding window, $\Delta T = 10$ ms. This resulted in an effective sampling rate of 100 Hz for the output cardiac signal. Let us denote the $i$th block on the $j$th microphone as $y^{(i,j)}(t)$. Performing a DFT over this signal gives us

$$Y^{(i,j)}(f) = \sum_{t=0}^{T} y^{(i,j)}(t) e^{-j2\pi ft/T} \qquad (2)$$

We next performed equalization to transform the received FMCW chirp into an impulse response. To do this, we canceled out the phase of the FMCW chirp, $\phi(f)$, in the frequency domain. Since the sliding window resulted in a timing synchronization offset, $i\Delta T \text{mod} T$, in the FMCW signal, it introduced an additional phase offset in the frequency domain, $-2\pi f \frac{\Delta T}{T} i$. We performed frequency domain equalization to cancel both these phases to obtain

$$\Psi^{(i,j)}(f) = e^{-j\phi(f) + j2\pi f \frac{\Delta T}{T} i} Y^{(i,j)}(f) \qquad (3)$$

The time-domain impulse response of the acoustic channel was then obtained by performing an inverse DFT to obtain

$$\psi^{(i,j)}(t) = \sum_{f=f_0 T}^{(f_0 + F)T} e^{j2\pi ft/T} \Psi^{(i,j)}(f) \qquad (4)$$

This impulse response represents the time of arrival of the various reflections from the speaker to the microphone.

Since cardiac motion is minute, it can be drowned out by reflections corresponding to coarse motion from distant locations. Therefore, we performed echo suppression to eliminate the reflections arriving from the farther distances. The impulse response at time $t$ represents the total energy of the reflections that arrive at time $t$. To reduce the effect of reflections from distant motion, we can zero out the impulse responses at farther distances. Since our operational range was $D = 1$ m, the round-trip time of arrival corresponding to this distance was $T_d = 2D/c$, where $c$ is the speed of sound. Zeroing the signal after $T_d$ in the impulse responses can lead to abrupt changes in the time domain and spectrum leakage in the frequency domain. Instead, we pointwise multiplied $\psi^{(i,j)}(t)$ with a raised-cosine window $W(t)$ starting at time 0, with a roll-off factor of 1 and length $T_d$. This yielded the impulse response after multipath suppression

$$\hat{\psi}^{(i,j)}(t) = \psi^{(i,j)}(t) W(t - T_d/2) \qquad (5)$$

We then performed a DFT on this impulse response to obtain $\hat{\Psi}^{(i,j)}(f)$.

*Adaptive maximum-SINR beamformer*. To motivate the need for an adaptive beamformer, we must understand how breathing motion interferes with the minute heart motion. The received acoustic signal at each microphone is a superposition of reflections from various reflectors on the body, including the chest, abdomen, and neck as well as reflections from static objects and noise. Assuming that breathing

and heartbeats result in a displacement of ~0.5 cm and 0.5 mm, respectively, this results in a phase change of around 3.3 and a 0.3 radian in the acoustic signal. Thus, the received acoustic signal in the complex domain can be represented as a linear combination of complex numbers corresponding to two arcs, the respiration arc, and the heartbeat arc, in addition to a constant complex offset from static reflections and noise.

The complex numbers corresponding to the respiration arc have a repeating motion along the arc, with a quasi-static respiration frequency ($R_{\text{resp}}$) of less than 20 cycles per minute (CPM) in adult humans. Projecting an ideal breathing signal onto the real and imaginary components results in sinusoidal waves. However, the breathing motion is not perfectly sinusoidal. As a result, while the majority of breathing energy in the frequency domain is at $R_{\text{resp}}$ and its second harmonic (<40 CPM), a nonnegligible portion of energy leaks into the higher frequencies that correspond to heart motion.

A heartbeat arc in comparison is much smaller, and the moving trajectory along each heartbeat arc can thus be approximated as a linear segment. Hence, the projection of the motion along the arc onto the real or imaginary axis is approximately linear to the motion itself. Human heartbeat motion has a mean frequency ($R_{\text{heart}}$) between 60 and 150 CPM. However, the instantaneous heart rate, which is the reciprocal of the R–R interval, is not necessarily quasi-static.

Without loss of generality, we can model the motion along the heartbeat arc as a carrier wave at a frequency $R_{\text{heart}}$ that is FM with a finite random signal $s(t)$ that changes the beat-to-beat interval. Since heartbeats have an average frequency of $R_{\text{heart}}$, the modulating signal $s(t)$ had a maximum bandwidth of $B = R_{\text{heart}}/2$. The FM modulation signal can then be written as

$$\text{FM}(t) = \cos\left(2\pi R_{\text{heart}}t + \delta f \int_0^t s(\tau)d\tau\right) \tag{6}$$

Here, $\Delta f$ is FM frequency deviation. The main assumption we make is that variations in beat-to-beat intervals have a maximum frequency such that $\Delta f < R_{\text{heart}}/2$. As a result, the modulated signal has a low modulation index as $\frac{\Delta f}{B} < 1$ and is a narrow-band FM signal. Given Carson's rule[54], the spectrum of narrow-band FM signals has only one main lobe, and the majority of the energy of the FM signal falls inside $R_{\text{heart}} \pm B$. Further, the spectrum has a long tail that is spread into frequencies outside this range.

The preceding analysis demonstrates two main properties of breathing and heart motion signals. First, a nonnegligible minority of the energy corresponding to breathing and heart motion can leak between these frequency ranges. Since the respiration motion is much larger than heartbeat motion, it introduces noise in the 60–150 CPM frequencies and can hide the heartbeat signal. As a result, band-pass filtering does not help to extract heart rhythm from the active sonar signal. Instead, we must design a beamforming algorithm. Second, most of the energy corresponding to breathing and heart motion falls in nonoverlapping frequencies of [0, 40] and [60, 150] CPM, respectively.

We leveraged both properties in the design of our maximum SINR beamformer. Taking 30 s of blocks as training sequences, the beamformer combined the signal across different microphones and frequencies in the impulse response to maximize the heart signal while minimizing the breathing signal and noise (see Supplementary Fig. 1). The frequency domain impulse response computed over the $i$th block and $j$th microphone can be written as

$$\hat{\Psi}^{(i,j)}(f) = \alpha_{j,f} S_i^{(\text{resp})} + \beta_{j,f} S_i^{(\text{heart})} + C_{j,f} + N_{i,j,f} \tag{7}$$

Here, $S_i^{(\text{resp})}$ and $S_i^{(\text{heart})}$ correspond to the respiration and heart signal, $\alpha$ and $\beta$ are the corresponding weights, $C_{j,f}$ corresponds to the reflections from the static objects in the environment, and $N$ is the noise. At a high level, the optimization problem aims to find the matrix $H = [h_{j,f}]$ such that $\frac{\sum_i |(H \cdot \beta) S_i^{(\text{heart})}|^2}{\sum_i |(H \cdot \alpha) S_i^{(\text{resp})}|^2 + \text{Var}(H \cdot N)}$ is maximized, where $A \cdot B = \sum_{i,j} A_{i,j} B_{i,j}$ and $\text{Var}(\cdot)$ denotes the variance.

The structure of respiration and heart signals is unknown since it varies across people and time. From the preceding analysis, the majority of the energy corresponding to breathing and heart motion lie in nonoverlapping frequencies. So, we instead used the energy in these frequency ranges as a proxy for breathing and heart motion in the above optimization. Specifically, we denote $S(i) = H \cdot \hat{\Psi}^{(i,j)}(f)$. We designed three FIR filters: a low-pass filter $W_{\text{resp}}$ with a cut-off frequency at 50 CPM, a band-pass filter $W_{\text{heart}}$ with a pass band of 60–150 CPM, and a high-pass filter $W_{\text{noise}}$ with a cut-off frequency at 150 CPM. We then computed the filtered signals as

$$\hat{S}_{\text{resp}} = W_{\text{resp}} * S, \hat{S}_{\text{heart}} = W_{\text{heart}} * S, \hat{S}_{\text{noise}} = W_{\text{noise}} * S \tag{8}$$

Here, * is the convolution operation. We then used gradient ascent to maximize the following objective function:

$$\mathcal{L}(H) = \log\left(||\Re(\hat{S}_{\text{heart}})||_2^2 + ||\Im(\hat{S}_{\text{heart}})||_2^2 + k\Re(\hat{S}_{\text{heart}}) \cdot \Im(\hat{S}_{\text{heart}})\right) - \log\left(\hat{S}_{\text{resp}}\hat{S}_{\text{resp}}^* + \hat{S}_{\text{noise}}\hat{S}_{\text{noise}}^*\right) \tag{9}$$

Here, $||A||_2$ is the 2-norm function of vector $A$, $\Re(\cdot)$ and $\Im(\cdot)$ represent the real and imaginary part of a complex number, and $S^*$ denotes the conjugate of $S$. We also

used a hyperparameter $k$ that constrained the level of coherence of the real (in-phase) and imaginary (quadrature) parts of the heart signal, because they were both linear projections of the same heart motion and hence should have a large correlation. Note that although we used a band-pass filter here, it was not used directly for signal extraction but only as a metric for approximating the SINR. After computing $H$ using gradient ascent, we extracted the heart rhythm signal $\hat{S}_{\text{heart}}$.

*Dropout and Regularization.* To avoid local maximum, we introduced two techniques during optimization. When random noise in any frequency-microphone pair has dominant energy within the heart rate range, it may be wrongly amplified while maximizing the objective function. We leveraged the fact that, unlike random noise, heartbeat motion should exist in a majority of frequency-microphones pairs. Hence, during the backward process in each iteration of gradient ascent, we probabilistically chose the weight to update with a probability $p = 0.6$, leaving the other weights unmodified.

The gradient ascent algorithm can also incorrectly converge to a local maximum that appears to be an impulse-like signal, which can be caused by a participant's abrupt motion. The length of the heartbeat arc, however, should not change abruptly over time because the skin displacement from each heartbeat is proportional to the blood pressure or apical impulse. Thus, the resulting signal should have a stable envelope. To enforce this, we introduced a regularization penalty term that is the maximum of the heart signal, i.e., $\max|\hat{S}_{\text{heart}}|$. Thus, the objective function we used in our gradient ascent algorithm is given by

$$\mathcal{L}(H) = -\log\left(||\Re(\hat{S}_{\text{heart}})||_2^2 + ||\Im(\hat{S}_{\text{heart}})||_2^2 + k\sum|\Re(\hat{S}_{\text{heart}})\Im(\hat{S}_{\text{heart}})|\right)$$
$$\log\left(\hat{S}_{\text{resp}}\hat{S}_{\text{resp}}^* + \hat{S}_{\text{noise}}\hat{S}_{\text{noise}}^* + \gamma\max(\hat{S}_{\text{heart}}\hat{S}_{\text{heart}}^*)\right) \tag{10}$$

We implemented the gradient ascent algorithm using PyTorch[55] with the parameters $k = 2$, $\gamma = 0.2$. The step size was initially set to 1, and we halved the step size if the objective function value did not increase every 100 iterations. Convergence was met when the step size fell below 0.05. The gradient ascent algorithm took an average of 2000 iterations to converge. The optimization was performed over the first 30 s of data to compute the beamforming matrix, $H$, which was then used to extract heart rhythms from the remaining data.

Finally, our algorithm does not use supervised learning in that it does not need ground truth data. Our optimization is self-supervised, which means that the inference for one person does not require ground truth training data for the person or pretrained model on other people. The self-supervised model extracts the hidden information (i.e., the R–R intervals) by optimizing the above objective function. The reason we use self-supervision is that different body shapes, positions, and the surrounding environments make a supervised model difficult to generalize. Instead, we identify the beamforming weights that maximize the signal strength of the heart rhythm motion by solving our optimization problem, without the need for any ground truth training data.

**Heartbeat segmentation.** After the beamforming process converged and $H$ was obtained, we extract the heart signal, $S_{\text{heart}}$, by applying a high-pass filter above 50 CPM to the real and imaginary parts of the resulting beamformed signal, $S$. We used a high-pass filter instead of a band-pass filter to preserve the high-frequency information and improve temporal resolution in the heartbeat signal.

We next segmented this complex signal into individual heartbeats. The challenge here is imperfect beamforming, which leaves residual interference from respiratory motion that modulates the heart signal. This introduces a rotation to the heartbeat signal, which changes the projection ratio between the real and imaginary components. Thus, we cannot always observe heartbeats only on the real (in-phase) or imaginary (quadrature) components (Fig. 2). Choosing local peaks from the absolute values of $S_{\text{heart}}$ does not work since the residual noise from the high-pass filter creates fake peaks; a more restrictive band-pass filter could reduce this noise but would also reduce temporal resolution.

We designed a segmentation algorithm that finds both the segmenting points and the rotation of each segment simultaneously. Our intuition was that the shapes of consequent heartbeat arcs were similar after accounting for temporal scaling due to different R–R intervals and a rotation between them due to residual breathing motion. The algorithm finds the segmenting point and the corresponding rotation transformation for each segment, where one segment post rotation is most similar to its previous segment after scaling to be the same duration. Unlike prior segmentation approaches[23,46], our algorithm is noniterative, accounts for rotations, and relies on comparison only between adjacent segments.

To measure the distance metric between segments $s_i$ and $s_{i+1}$, we first normalized their lengths to the longer segment using linear interpolation (see Supplementary Algorithm 1). The best rotation was then computed by minimizing the mean square error between $s_i$ and the rotated $s_{i+1}$. This rotation is given by

$$s_{i+1}^{(rot)} = s_{i+1}\sqrt{\frac{s_i s_{i+1}^*}{s_{i+1} s_i^*}} \tag{11}$$

Given two complex vectors $x$ and $y$ with $L$ elements each, the rotation angle, $\theta$, that

minimizes the mean square error

$$E = \sum_{i=1}^{L} (x_i \exp(j\theta) - y_i)(x_i \exp(j\theta) - y_i)^* = \sum_{i=1}^{L} x_i x_i^* - x_i y_i^* \exp(j\theta) - x_i^* y_i \exp(-j\theta) + y_i y_i^* \tag{12}$$

This can be computed by setting the first derivative to 0 as follows:

$$\frac{dE}{d\theta} = \sum_{i=1}^{L} -jx_i y_i^* \exp(j\theta) + jx_i^* y_i \exp(-j\theta) = 0 \tag{13}$$

Thus, an optimal rotation is given by

$$\exp(j\theta) = \sqrt{\frac{x^* y}{y^* x}} \tag{14}$$

The distance metric between two segments was then defined as

$$d(s_i, s_{i+1}) = \frac{||s_i - s_{i+1}^{(rot)}||_2^2}{||s_i + s_{i+1}^{(rot)}||_2^2} \tag{15}$$

Once we identified each beat segment, we chose its midpoint as the timing for the corresponding heartbeat, which we then used to compute the heart rate and R–R intervals.

**Synchronizing different data streams**. To compare the heart rate and R–R intervals computed by our algorithm to the ground truth from the ECG and PPG sensors, we needed to synchronize both data streams and match their corresponding heartbeats. We first corrected the initial timing offsets using two steps. Initially, we started the sensor tracking on the smartphone ~5 s after we turned on the acoustic signal recording using a manual timer. Then, before processing, we offset each acoustic recording by 5 s to achieve a coarse synchronization with the ground truth. To accurately match the start timings, the alignments were manually examined and adjusted to match the first heartbeat across data streams. The timing of each beat was extracted from the acoustic recordings using our algorithms, and the heart rate was calculated by counting the number of beats within 1 min. The manual alignment is carefully performed to match the first heartbeats to minimize errors for the remaining heartbeats in each data stream.

Another well-known challenge encountered when comparing R–R intervals across data streams is that any missed heartbeat in one of the data streams can affect all subsequent R–R intervals since synchronization is lost; this results in our comparing R–R intervals across data streams that are not synchronized with each other[56,57]. To perform this matching across the ground truth annotations of the heartbeats and our algorithm output, we first matched each R–R interval segment for both data streams. Say, $t_i$ and $t_i'$ are beat timings in ground truth annotations and our algorithm output, respectively. For each beat $i$ in the ground truth annotations, we find the beat $f(i)$ in the algorithm output where $|t_i - t_{f(i)}'|$ is the smallest. Similarly, for each beat $j$ in the algorithm output, we find the $g(j)$ in the ground truth annotation where $|t_{g(j)} - t_j'|$ is the smallest. We matched R–R intervals where starting and ending beats mutually matched each other across the two streams, i.e., where $g(f(i)) = i$ and $g(f(i+1)) = i+1$, and no other heartbeats matched the beats in the R–R intervals. Using this matching process, 86.7% of R–R intervals were matched across healthy participants and cardiac patients. This is a similar fraction to that reported in prior work comparing R–R intervals between Apple watch and the gold standard ECG[56]. Excluding unmatched R–R intervals, however, might lead to more optimistic results since the mismatch is likely due to poor signal quality. To understand the effect of this exclusion, we included all the R–R intervals for the healthy cohort and compared the two data streams using the interpolation method in ref. [58]. The excluded intervals above, follow the error types 4 and 5 in ref. [58], where M intervals in our results correspond to N ≠ M intervals in the ground truth. We interpolated them into max(M,N) intervals evenly. This increased the absolute median error from 28 to 32 ms and the 90th percentile error from 75 to 89 ms.

**Statistics and reproducibility**. We analyzed heart rate and R–R intervals using standard statistical methods using Python (Python Software Foundation, Delaware, USA) and the figures were generated using Python Matplotlib library. We computed the bias error (mean of the errors), precision error (standard deviation of the errors), MAE (median value of the absolute errors), and 90th percentile error (value below which 90% of the absolute errors fall when sorted). We plotted all data points in scatter plots and showed the limits of agreement as two times the standard deviation. The ICC and CCC were calculated using the equations outlined in refs. [59] and [60], respectively.

For clinical variables of interest, continuous variables were reported as the mean ± standard deviation, and categorical variables were reported as the number (percentage). Statistical analysis was performed using SAS University Edition (SAS Institute, Cary, NC, USA).

**Reporting summary**. Further information on research design is available in the Nature Research Reporting Summary linked to this article.

## Data availability

All data required to interpret the results are included in the manuscript. Supplementary Data 1–3 include raw data points to produce Figs. 3–5.

## Code availability

We wrote custom C++ code using generic audio drivers for data collection using our customized smart speaker prototypes. We wrote custom Python code for signal processing using common open-source libraries such as NumPy and PyTorch. Code for the data collection and signal processing is available upon request with a noncommercial license.

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

## Acknowledgements

The authors would like to thank our participants and their families at the University of Washington Medical Center for their willingness to participate in this study. The authors are grateful for funding from the National Science Foundation. The authors thank Dr. Kamal Shah and Thurman Gillespy for their insights on device assembly, setup, and testing, and Sandy Kaplan, Justin Chan, Dr. Kelly Michaelsen, Dr. Jacob Sunshine, Vicente Arroyos, and Ali Najafi for their critical feedback on the manuscript.

## Author contributions

All authors designed the experiments; A.W. and D.N. conducted the experiments; A.W. developed software and deployed the algorithm; A.W. and D.N. conducted the analysis; all authors interpreted results; A.W. generated figures; and all authors wrote the manuscript. Conceptualization: S.G. and A.W.

## Competing interests

All authors are inventors in the provisional patent application in the process of being submitted by the University of Washington; A.W. and S.G. have equity stakes in Sound Life Sciences, Inc.; and S.G. is a cofounder of Jeeva Wireless, Inc., Sound Life Sciences, Inc., and Wavely Diagnostics, Inc.
