## [Peer Review File · Communications Biology]

Reviewers' comments:

Reviewer #1 (Remarks to the Author):

The paper describes an approach which uses acoustic signals from commodity devices to measure heart rhythm features. The advantages of such an idea cannot be overrated: especially with current pandemic, the ability to have a contactless approach is very valuable. I think the paper could also try to make stronger case for the use of the approach in a self administered test, which could allow remote diagnosis. Perhaps this could be framed more strongly in the introduction.

While one might say that the specific technical contribution is very similar to 25 and 36, the adaptation of the techniques to frequency and sampling which is compatible to commodity smart home speakers is of value. The most important contribution here is the extensive evaluation with healthy individuals and heart patients.

In the results description of the cardiac patients, it would be nice to have more details on "how good" the results are: what does it mean to have that sort of error in practice? Would this compromise the diagnosis? In general more comments/interpretation over the meaning of the results would be good.

The methods section explains very well the details of the approach: I am left with a question on why these specific frequency ranges are used. It is mentioned that the frequencies used are 18-22kHz which are audible to younger humans and these while other smart speakers are at 25-23 kHz. The justification of this choice should be given. Also speculation on what would work in this frequency range if the algorithm was to be applied would be good: ideally experiments would be good but I think it might be beyond what is possible to do.

It would be good to check what is the audible spectrum for pets — for example dogs and cats can hear higher frequencies than people, so would be great to see whether the proposed 25-30 kHz falls above that. Also, increased frequency would definitely require further testing — it might reduce the operational distance from 1 meter, and might make the system even more sensitive to thick layers of clothing or signal attenuation through adipose tissue.

The discussion section does a good job in describing the limitations of the work: I praise the authors for such a sincere report.

I have some further minor comments:

- When it comes to testing the system with the background music playing, it is not mentioned what is the volume of the music playing.
- some error on the hospitalised cardiac patients are reported. It would be interesting to see some discussion on what this error could mean in terms of diagnosing patients with arrhythmia. Single miscalculated R-R interval for one patient shouldn't make much of a difference, with the arrhythmia then spotted in the subsequent R-R intervals. But if the error is repeated for many R-R intervals for the same patient, it would lead to incorrect diagnosis. Same goes for healthy patients. Would be great to see whether the errors are random, or whether some patients have more errors than others (if so, are there any metrics that could be related to the poorer performance on specific patients?)
- Any idea why median error for female participants higher than for males (30 vs 27 ms)? Most likely due to females having more fatty tissue in the chest area. A good idea could be to look on the non-obese population whether there is any correlation between BMI and error on the R-R interval.
- Unsure whether it is OK to use two different data collection methods for ground truth (namely, ECG for some patients, and PPG for patients who couldn't get ECG). While I understand the rationale behind this, would be good to see a little more insight into how the data differs by collecting both ECG and PPG from the patients who could get the ECG.
- It would be great not only to see a discussion on the implications of the limitations of the

proposed measurement method, but also on the potential ways of overcoming these in the future (not just for the frequency spectrum, but also for movement, multiple people present, the requirement for the user to sit directly across and the speaker to be set up on the nipple level, etc.)

- The performance of this methodology is compared to Apple Watch, which is a PPG (i.e. relying on contact) methodology. It would be also good to see the comparison with other contact-free methods, microwave approaches, etc.

Reviewer #2 (Remarks to the Author):

The authors of the paper present a novel approach for contactless monitoring of cardiac activity using a smart speaker by turning it into an active sonar system. They create frequency-modulated signals from 18-22 kHz to utilize the FMCW principle in order to gather vital sign data from various participants. Overall, the authors have done a comprehensive research on this topic. The article is well structured and the experiments are well conducted. What is also positive is the fact that they not only measured healthy participants but also hospitalized patients. There are a few questions to the authors remaining after reading the article.

Figure 6 G/H: It surprises me to see no variation of the heart rate at all in the ECG data of G and H. Even with healthy people there should be some variation (hence the heart rate variation in healthy people), e.g., due to respiration as shown in F. Do you know the possible reason for it? Might an (cardiac) illness of the persons be the reason for this?

In Discussion you mention several limitations. Another limitation that I see would be that it might be difficult to perform long-term monitoring since the person needs to be sitting still in front of the smart speaker. Are there any feasible approaches for long-term monitoring, also for example at nighttime? Otherwise, I would mention this limitation.

Have you tried measuring at the side and/or the back of a person? Would this still work? This would give your system another significant degree of freedom.

Line 246-250: You mention that your waves are generally inaudible to adults. However, I thought of animals/pets like dogs or cats, which can clearly hear these sounds. Do you know if these frequency ranges would cause discomfort for them? This would be another (small) limitation I guess?

Concerning your measurements/signal processing: Did you by chance evaluate your minimal distance resolution that you can still measure? E.g., if a vibration with say 0.1mm is still measurable.

Adaptive maximum-SINR beamformer: You mentioned that you use 30s blocks as training sequences. Combining this with what you have written in lines 408-410, does that mean that you optimized your algorithm on the data from the person that you test it on? If so, I would suggest including some kind of cross-correlation, i.e., using different persons for training H and testing.

Line 447-450: Do I understand correctly that you synchronize your data streams by synchronizing the ECG RR-signal and the RR-signal that you get from your acoustic signal using your algorithm? Did you evaluate the accuracy you can achieve using this approach? I image it to be difficult to determine the exact delay since the start of the ECG heartbeat (R-peak) does not exactly correspond to the peak from your algorithm. The peak from your acoustic signal is delayed since you measure a mechanical reaction, however, depending on your measuring position etc., the delay might vary.

Line 462: You mention that 86.7% of RR intervals were matched. What happened to the RR intervals that you could not match? Did you include them in your analysis that you present in "Results"? If not, you should definitely include them somehow; else, you would optimize your results using reference ECG information.

Reviewer #3 (Remarks to the Author):

The ability to monitor heart beats using smart speakers is very useful. It is very challenging since heart beats cause very tiny movements. There have been several previous works including those cited by the authors that can monitor breathing using smart speakers. But heart beats lead to even smaller movements, and it is surprising smart speakers can pick up the movements. The user studies across different people including the effect of obesity is interesting.

The authors could elaborate on the details for the beamforming design. Their description suggested they used DNN. What is the DNN structure? How much training data is required to train the DNN? How well the DNN generalizes?

Some heart patients have heart beats below 60 CPM. How does the current design handle this case?

Clothes have significant impacts. Loose clothes even if thin can significantly reduce the accuracy. Do you see that?

There is a closely related paper and it'd be good to compare with:

Contactless Seismocardiography via Deep Learning Radars. Unsoo Ha, Salah Assana (MIT Media Lab); Fadel Adib (Massachusetts Institute of Technology). MobiCom 2020.

We thank the reviewers for all of their insightful comments and helpful suggestions. We have carefully reviewed the comments and made significant clarifications and revisions to the manuscript to address the concerns raised by the reviewers. We also reported additional experimental results to further evaluate our proposed system. Below we list the responses to each comment in detail. We believe the manuscript has become stronger as a result of your feedback. Thank you for your time.

Reviewer 1:

The paper describes an approach which uses acoustic signals from commodity devices to measure heart rhythm features. The advantages of such an idea cannot be overrated: especially with current pandemic, the ability to have a contactless approach is very valuable. I think the paper could also try to make stronger case for the use of the approach in a self administered test, which could allow remote diagnosis. Perhaps this could be framed more strongly in the introduction.

Thank you very much for this suggestion. We have added the following in lines 35-40: "Contactless rhythm acquisition may also be valuable in the modern telemedicine era, whereby patients can be trained for relatively simple self-administered rhythm analysis, and this data could be then transmitted to their physician. The benefits of a self-administered test are numerous; and may include the ability to connect patients living in rural areas to physicians and clinical investigators, screen patients for atrial fibrillation and heart rhythm disorders remotely, and obtain clinical trial data without the need for an in-person visit."

While one might say that the specific technical contribution is very similar to 25 and 36, the adaptation of the techniques to frequency and sampling which is compatible to commodity smart home speakers is of value. The most important contribution here is the extensive evaluation with healthy individuals and heart patients.

Thanks for your positive comments on our contribution.

In the results description of the cardiac patients, it would be nice to have more details on "how good" the results are: what does it mean to have that sort of error in practice? Would this compromise the diagnosis? In general more comments/interpretation over the meaning of the results would be good.

We thank the reviewer for their comment. To address this comment, we added the following to lines 202-207: "Within the context of clinical practice, it is unlikely that this magnitude of error would result in diagnostic errors for detecting atrial fibrillation where R-R interval variation less than 50 ms is often not clinically significant. In atrial fibrillation, the R-R interval widely varies from beat to beat and standard deviations range between 95-233 ms in different physiological states (29). Proper diagnosis of rhythm disorders relies on the ability to detect temporally disparate R-R intervals, rather than precise R-R interval measurement."

The methods section explains very well the details of the approach: I am left with a question on why these specific frequency ranges are used. It is mentioned that the frequencies used are 18-22kHz which are audible to younger humans and these while other smart speakers are at 25-23 kHz. The justification of this choice should be given. Also speculation on what would work in this frequency range if the algorithm was to be applied would be good: ideally experiments would be good but I think it might be beyond what is possible to do. It would be good to check what is the audible spectrum for pets — for example dogs and cats can hear higher frequencies than people, so would be great to see whether the proposed 25-30 kHz falls above that. Also, increased frequency would definitely require further testing — it might reduce the operational distance from 1 meter, and might make the system even more sensitive to thick layers of clothing or signal attenuation through adipose tissue. The discussion section does a good job in describing the limitations of the work: I praise the authors for such a sincere report.

This is a good question. The frequency range we use is not audible for most adults, while children are able to hear up to 22 kHz and pets like dogs and cats are even able to hear frequencies up to 40 kHz. One of the reasons we do not use higher frequencies is the hardware limitation. The audio input and output of our prototype device are limited to a 48 kHz sampling rate, which translates to a frequency of 24 kHz. Further the frequency response of our hardware significantly degrades beyond 22 kHz. Commercial smart speakers such as Google Nest potentially support a higher sampling rate and hence commercial Nest products use ultrasound sensing at 25-30 kHz in millions of their smart speaker devices. Generally speaking, higher frequencies are associated with higher attenuation rate. However, we believe that a slightly higher frequency range if the hardware permits (e.g., 25-30kHz) could benefit generalization across all age groups and some of our pets. We updated our discussion section to include the above information.

We also thank the reviewer for raising the concern about the potential effect for children and pets. For adults, the WHO recommends a noise limit of 85dB (A) on an average of 8 working hours [1]. Our exposure intensity (i.e., 75dB, which is approximately 66dB (A) at 20kHz, at 50cm) is much less than that. For infants, they are able to hear higher frequency sound. However, short-time exposure to high frequency does not affect their hearing capability nor health [2,5]. Pets like cats and dogs have even higher sensitivity to ultrasound as high as 64kHz [3,4]. There are research literatures which show that sound around 40kHz that cannot be heard by human beings can potentially interrupt the sleep of our pets [6] or cause feline audiogenic reflex seizures for cats [7]. However sounds in the 18-22~kHz are not known to affect animals. In our prior work using active sonar for breathing monitoring in self-injection facilities, some of the participants brought in their dogs, which did not have any observable reaction to the 18-22~kHz FMCW signals. We will add this into our discussion section.

Specifically, we added the following text in lines 287-298: "Our smartspeaker prototype has a sampling rate of 48 kHz and uses 18--22 kHz acoustic transmissions which are generally inaudible to adults but can be audible to the younger population. Commercial smart speakers like Google Nest support acoustic frequencies between 25--30 kHz, which are inaudible across the age spectrum and could be used to enable cardiac rhythm monitoring using our algorithms. Frequencies higher than 30 kHz require specialized hardware and also limits the range of the system. WHO recommends a noise limit of 85dB(A) over an average duration of 8 working hours. Our exposure intensity (i.e., 75dB, which is approximately 66dB(A) at 20kHz and 50cm) is much less than that. Short-time exposure to high frequency also does not affect the hearing capability of infants. Pets like dogs have even higher sensitivity to ultrasound as high as 64kHz and sound around 40 kHz can potentially interrupt their sleep and cause feline audiogenic reflex seizures for cats. However sounds in the 18-30 kHz are not known to affect animals. Prior active sonar studies report that 18--22 kHz FMCW signals did not elicit reaction from dogs."

[1] Goelzer, Berenice, Colin H. Hansen, and G. Sehrndt. Occupational exposure to noise: evaluation, prevention and control. World Health Organisation, 2001.

[2] Hanson, Mark A. Health effects of exposure to ultrasound and infrasound: Report of the independent advisory group on non-ionising radiation. Health Protection Agency, 2010.

[3] Fay, Richard R., and Laura Ann Wilber. "Hearing in vertebrates: a psychophysics databook." (1989): 2044-2044.

[4] Gay, W. I. Methods of animal experimentation. Volume IV. Environment and the special senses. 1973. pp 43-143.

[5] Repacholi, Michael H. "Ultrasound: Characteristics and biological action." (1981).

[6] Turner, Jeremy G., et al. "Hearing in laboratory animals: strain differences and nonauditory effects of noise." *Comparative medicine* 55.1 (2005): 12-23.

[7] Lowrie, Mark, et al. "Audiogenic reflex seizures in cats." *Journal of feline medicine and surgery* 18.4 (2016): 328-336.

I have some further minor comments:

- When it comes to testing the system with the background music playing, it is not mentioned what is the volume of the music playing.

We thank the reviewer for pointing it out. The music was played at around 75 dBA. We updated this is line 132. Thank you.

- some error on the hospitalised cardiac patients are reported. It would be interesting to see some discussion on what this error could mean in terms of diagnosing patients with arrhythmia. Single miscalculated R-R interval for one patient shouldn't make much of a difference, with the arrhythmia then spotted in the subsequent R-R intervals. But if the error is repeated for many R-R intervals for the same patient, it would lead to incorrect diagnosis. Same goes for healthy patients. Would be great to see whether the errors are random, or whether some patients have more errors than others (if so, are there any metrics that could be related to the poorer performance on specific patients?)

Thank you very much for this comment. This was an intriguing question and so we plotted the absolute median and 90-percentile R-R interval errors amongst the hospitalized population.

The graph shows that there is no specific trend, except that higher median R-R intervals result in higher 90-percentile error. We did not observe significant decrease in accuracy among those with irregular rhythms compared to those with regular rhythms.

We added this graph as Supplementary Figure 4 and discussed this in lines 200-202: "Higher median R-R intervals resulted in higher 90-percentile error (Supplementary Figure 4). There was no significant decrease in accuracy among those with irregular rhythms compared to those with regular rhythms."

As noted above, this error rate is unlikely to be clinically significant in terms of arrhythmia diagnosis even in irregularly irregular rhythms such as atrial fibrillation.

- Any idea why median error for female participants higher than for males (30 vs 27 ms)? Most likely due to females having more fatty tissue in the chest area. A good idea could be to look on the non-obese population whether there is any correlation between BMI and error on the R-R interval.

Thank you for this great idea. Since the healthy participants have a range of BMIs in the non-obese range, we have included a new graph (new Supplementary Figure 3) that shows the correlation between BMI and error in the R-R intervals. We included both the median and the 90-percentile errors for each of the participants. The figure shown below shows the errors increase slightly with BMI.

The slightly higher error for female participants is likely because of adipose and breast tissue which attenuates the motion of the heart at the chest surface. We clarified this in lines 228-233.

- Unsure whether it is OK to use two different data collection methods for ground truth (namely, ECG for some patients, and PPG for patients who couldn't get ECG). While I understand the rationale behind this, would be good to see a little more insight into how the data differs by collecting both ECG and PPG from the patients who could get the ECG.

Thank you for pointing this out. In general, PPG sensors that we use provide a RR interval with an accuracy comparable to an ECG sensor with correlation coefficients between 0.998 and 0.968 in prior work cited below. To verify, we also did a comparison test between the two sensors we use (EliteHRV Corsense PPG sensor, Polar H10 ECG sensor) on two healthy participants. We achieved a mean absolute difference of 11ms. Thus, the measurement error is smaller compared to the error using our proposed system. We will report this in the discussion section.

Selvaraj, Nandakumar, et al. "Assessment of heart rate variability derived from finger-tip photoplethysmography as compared to electrocardiography." *Journal of medical engineering & technology* 32.6 (2008): 479-484.

Liu, Weichao, et al. "Reliability analysis of an integrated device of ECG, PPG and pressure pulse wave for cardiovascular disease." *Microelectronics Reliability* 87 (2018): 183-187.

We added 333-337 lines to address this comment: "In the study, we use the EliteHRV Corsense PPG and Polar H10 ECG sensors for ground truth. PPG sensors are known to produce comparable R-R interval accuracies to ECG, with high correlation coefficients between 0.968 and 0.998 [51,52]. To verify this, we performed a comparison test between the ground truth sensors on two healthy participants and noted that the mean absolute R-R interval difference was 11 ms."

- It would be great not only to see a discussion on the implications of the limitations of the proposed measurement method, but also on the potential ways of overcoming these in the future (not just for the frequency spectrum, but also for movement, multiple people present, the requirement for the user to sit directly across and the speaker to be set up on the nipple level, etc.)

We thank the reviewer for this suggestion. We added the following text to lines 278-286: "At this time, our system is designed for spot monitoring of a single participant. Further hardware and software enhancements could enable

continuous monitoring. To improve signal strength and range, the smart speaker hardware may need directional tweeter which can rotate to the direction of interest as well as speakers with a better response at the target frequencies and microphones with higher sampling rates and bit resolutions. New smart speaker models have rotatable directional tweeter capabilities (e.g., Amazon Echo Show 10) and have microphones and speakers that are designed to operate at the target frequencies; in contrast our hardware has a 10-15 dB degradation at 18-22 kHz. Multiple participants could be supported using FMCW algorithms that use breathing motion to track the location of each participant and separating the signals arriving from different distances (20).”

- The performance of this methodology is compared to Apple Watch, which is a PPG (i.e. relying on contact) methodology. It would be also good to see the comparison with other contact-free methods, microwave approaches, etc.

Thank you for this comment. To address this, we added lines 299-303 to the discussion section: “Radar-based systems use radio signals with large bandwidths and use custom hardware that is not pervasive in smart speakers. Prior radar-based studies report a median error in R-R intervals of 8-44 ms for healthy participants (13,46-48) and 186 ms for cardiac patients with atrial fibrillation (10). Our sound-based system instead uses active sonar algorithms, hardware that is pervasive in smart speakers and is designed to achieve low errors for both regular and irregular rhythm.”

Reviewer 2:

The authors of the paper present a novel approach for contactless monitoring of cardiac activity using a smart speaker by turning it into an active sonar system. They create frequency-modulated signals from 18-22 kHz to utilize the FMCW principle in order to gather vital sign data from various participants. Overall, the authors have done a comprehensive research on this topic. The article is well structured and the experiments are well conducted. What is also positive is the fact that they not only measured healthy participants but also hospitalized patients. There are a few questions to the authors remaining after reading the article.

We thank the reviewer for the positive comment and suggestions.

Figure 6 G/H: It surprises me to see no variation of the heart rate at all in the ECG data of G and H. Even with healthy people there should be some variation (hence the heart rate variation in healthy people), e.g., due to respiration as shown in F. Do you know the possible reason for it? Might an (cardiac) illness of the persons be the reason for this?

We thank the reviewer for the astute observation. We provided the following clarification in lines 212-224: “Figure 6 G corresponds to a patient with an implanted permanent cardiac pacemaker and a paced rhythm. The patient in Figure 6 H had an intrinsic rhythm (non-paced rhythm); and this patient had mild variations in the R-R intervals with a standard deviation less than 10ms. This low level of heart rate variability is not uncommon. We collected data from patients in the cardiac floor of our tertiary care medical center with a variety of cardiac conditions, which included cardiac conduction disorders, arrhythmias, cardiomyopathy as well as valvular disorders. Many of these cardiac conditions directly or indirectly affect the heart rate variability. Respiratory sinus arrhythmia, which is a major cause of heart rate variability becomes less common with age (30) and is less prevalent in patients with diabetes due to autonomic neuropathy (31). Our hospitalized population had a mean age of 63.2 years in the regular rhythm group and 68.0 in the irregular rhythm group, and there were a total of 5 out of 24 patients with diabetes. In addition, medications that influence vagal tone, such as beta blockers, digoxin, opiate pain medications may decrease sinus

arrhythmia. (32). Our sample of hospitalized cardiac patients often had multiple factors which could reduce heart rate variability. (Figure 6G,H). ”

[30] Kaushal, Padmini, and J. Andrew Taylor. "Inter-relations among declines in arterial distensibility, baroreflex function and respiratory sinus arrhythmia." *Journal of the American College of Cardiology* 39.9 (2002): 1524-1530.

[31] Smith, Shirley A. "Reduced sinus arrhythmia in diabetic autonomic neuropathy: diagnostic value of an age-related normal range." *Br Med J (Clin Res Ed)* 285.6355 (1982): 1599-1601.

[32] Taylor, J. Andrew, et al. "Sympathetic restraint of respiratory sinus arrhythmia: implications for vagal-cardiac tone assessment in humans." *American Journal of Physiology-Heart and Circulatory Physiology* 280.6 (2001): H2804-H2814.

In Discussion you mention several limitations. Another limitation that I see would be that it might be difficult to perform long-term monitoring since the person needs to be sitting still in front of the smart speaker. Are there any feasible approaches for long-term monitoring, also for example at nighttime? Otherwise, I would mention this limitation.

We thank the reviewer for this suggestion. We added the following text to lines 278-286: “ At this time, our system is designed for spot monitoring of a single participant. Further hardware and software enhancements could enable continuous monitoring. To improve signal strength and range, the smart speaker hardware may need directional tweeter which can rotate to the direction of interest as well as speakers with a better response at the target frequencies and microphones with higher sampling rates and bit resolutions. New smart speaker models have directional tweeter capabilities (e.g., Amazon Echo Show 10) and have microphones and speakers that are designed to operate at the target frequencies; in contrast our hardware has a 10-15 dB degradation at 18-22 kHz. Multiple participants could be supported using FMCW algorithms that use breathing motion to track the location of each participant and then separate cardiac signals arriving from different distances (20).”

Have you tried measuring at the side and/or the back of a person? Would this still work? This would give your system another significant degree of freedom.

This is a great suggestion. We have done additional experiments with 5 healthy participants by placing the device towards the left side, right side, and back of the participant. We report the R-R interval error of them in the Supplementary Figure 2 (attached below). As we can see, while the device could compute the R-R intervals from the front, right and left, it does not work well when placed at the back of a person.

We added this figure and the following to lines 152-154: “The algorithm is resilient to larger angles with the smart speaker placed to the left and right of the participant; the error however is high when placed behind the participant facing their back (Supplementary Figure 2).”

Line 246-250: You mention that your waves are generally inaudible to adults. However, I thought of animals/pets like dogs or cats, which can clearly hear these sounds. Do you know if these frequency ranges would cause discomfort for them? This would be another (small) limitation I guess?

We thank the reviewer for raising the concern about the potential effect for children and pets. As mentioned earlier, for adults, the WHO recommends a noise limit of 85dB (A) on an average of 8 working hours [1]. Our exposure intensity (i.e., 75dB, which is approximately 66dB (A) at 20kHz, at 50cm) is much less than that. For infants, they are able to hear higher frequency sound. However, short-time exposure to high frequency does not affect their hearing capability nor health [2,5]. Pets like cats and dogs have even higher sensitivity to ultrasound as high as 64kHz [3,4]. There are research literatures which show that sound around 40kHz that cannot be heard by human beings can potentially interrupt the sleep of our pets [6] or cause feline audiogenic reflex seizures for cats [7]. However sounds in the 18-22~kHz are not known to affect animals. In our prior work using active sonar for breathing monitoring in self-injection facilities, some of the participants brought in their dogs, which did not have any observable reaction to the 18-22~kHz FMCW signals. We will add this into our discussion section.

Specifically, we added the following text in lines 287-298: “Our smartspeaker prototype has a limited sampling rate of 48 kHz and uses 18--22 kHz acoustic transmissions which are generally inaudible to adults but can be audible to the younger population. Commercial smart speakers like Google Nest support acoustic frequencies between 25--30 kHz, which are inaudible across the age spectrum and could be used to enable cardiac rhythm monitoring using our algorithms. Frequencies higher than 30 kHz require specialized hardware and also limits the range of the system. WHO recommends a noise limit of 85dB(A) over an average duration of 8 working hours. Our exposure intensity (i.e., 75dB, which is approximately 66dB(A) at 20kHz and 50cm) is much less than that. Short-time exposure to high frequency also does not affect the hearing capability of infants. Pets like dogs have even higher sensitivity to

ultrasound as high as 64kHz and sound around 40kHz can potentially interrupt their sleep and cause feline audiogenic reflex seizures for cats. However sounds in the 18-30 kHz are not known to affect animals. Prior active sonar studies report that 18--22 kHz FMCW signals did not elicit reaction from dogs."

[1] Goelzer, Berenice, Colin H. Hansen, and G. Sehrndt. Occupational exposure to noise: evaluation, prevention and control. World Health Organisation, 2001.

[2] Hanson, Mark A. Health effects of exposure to ultrasound and infrasound: Report of the independent advisory group on non-ionising radiation. Health Protection Agency, 2010.

[3] Fay, Richard R., and Laura Ann Wilber. "Hearing in vertebrates: a psychophysics databook." (1989): 2044-2044.

[4] Gay, W. I. Methods of animal experimentation. Volume IV. Environment and the special senses. 1973. pp 43-143.

[5] Repacholi, Michael H. "Ultrasound: Characteristics and biological action." (1981).

[6] Turner, Jeremy G., et al. "Hearing in laboratory animals: strain differences and nonauditory effects of noise." *Comparative medicine* 55.1 (2005): 12-23.

[7] Lowrie, Mark, et al. "Audiogenic reflex seizures in cats." *Journal of feline medicine and surgery* 18.4 (2016): 328-336.

Concerning your measurements/signal processing: Did you by chance evaluate your minimal distance resolution that you can still measure? E.g., if a vibration with say 0.1mm is still measurable.

We thank the reviewer for their comment. To address this comment, we added the following lines 350-356: "The minimum distance resolution achieved by our system depends on various factors that affect phase error: hardware components, circuit design and interference control, operating system and driver to support high-throughput audio signals, and the algorithm itself. The mean phase error on our acoustic hardware is approximately 0.05 radian in an empty room. Assuming signals from each of the 7 microphones are independent, the corresponding mean displacement error, with ideal beamforming, is around 0.025 mm. Note that this is an ideal distance resolution for our specific hardware and is likely better for consumer smart speakers with better hardware."

Adaptive maximum-SINR beamformer: You mentioned that you use 30s blocks as training sequences. Combining this with what you have written in lines 408-410, does that mean that you optimized your algorithm on the data from the person that you test it on? If so, I would suggest including some kind of cross-correlation, i.e., using different persons for training H and testing.

We apologize for the confusion: we are *not* performing training in the sense that one provides ground truth data in the training phase. We *do not* use any ground truth data for identifying the weights of our beamformer. Our optimization is self-supervised, which means that the inference for one person does not require ground truth training data for the person or pretrained model on other people. The self-supervised model extracts the hidden information (i.e., the RR intervals) by optimizing an objective function with constraints using the test data from itself. The reason we use self-supervision is that different body shape, position and the surrounding environment make a supervised model difficult to generalize. Since we do not use ground truth data for learning the weights and since the weights are different for each position/person, cross-correlation is not appropriate for our algorithm. We are identifying the beamforming weights that maximize the signal strength of the heart rhythm motion by solving our optimization problem, without the need for any ground truth training data. The analogy is traditional (per-neural networks) beamforming algorithms that use an optimization function to maximize SNR, without any ground truth training data.

To address this confusion, we added lines 475-482: "Finally, our algorithm does not use supervised learning in that it does not need ground truth data. Our optimization is self-supervised, which means that the inference for one person does not require ground truth training data for the person or pre-trained model on other people. The self-supervised model extracts the hidden information (i.e., the R-R intervals) by optimizing the above objective function. The reason we use self-supervision is that different body shapes, positions and the surrounding environments make a supervised

model difficult to generalize. Instead, we identify the beamforming weights that maximize the signal strength of the heart rhythm motion by solving our optimization problem, without the need for any ground truth training data.”

Line 447-450: Do I understand correctly that you synchronize your data streams by synchronizing the ECG RR-signal and the RR-signal that you get from your acoustic signal using your algorithm? Did you evaluate the accuracy you can achieve using this approach? I image it to be difficult to determine the exact delay since the start of the ECG heartbeat (R-peak) does not exactly correspond to the peak from your algorithm. The peak from your acoustic signal is delayed since you measure a mechanical reaction, however, depending on your measuring position etc., the delay might vary.

Thank you for pointing out. It is indeed difficult to align the ECG RR-signal with our RR-signal due to differences in the two devices, and the timing difference between electric signal and mechanical reaction. Hence, we carefully and manually align the starting point of the two sequences for each data stream. While time consuming and tedious, this was essential to compute the error and compare the two data streams.

We emphasize this in lines 519-524: “To accurately match the start timings, the alignments were manually examined and adjusted to match the first heart beat across data streams. The timing of each beat was extracted from the acoustic recordings using our algorithms, and the heart rate was calculated by counting the number of beats within one minute. The manual alignment is carefully performed to match the first heart beats to minimize errors in the rest of heart beats in each data stream.”

Line 462: You mention that 86.7% of RR intervals were matched. What happened to the RR intervals that you could not match? Did you include them in your analysis that you present in "Results"? If not, you should definitely include them somehow; else, you would optimize your results using reference ECG information.

We did not include the unmatched RR intervals in the results section, because there is no one-to-one RR-interval mappings between the intervals from the ground truth ECG device and our system in those periods with heart rate errors. For a relatively fair comparison, we exclude them when we interpret our results.

We thank the reviewer to point out that the exclusion of them optimizes our results since the mismatch is likely due to poor SNR, and in which cases, the RR-interval estimation would likely have a higher error assuming a perfect segmentation were applied. It is however difficult to remove the bias without using the information from the groundtruth. As suggested, we have now also included the errors that include all the R-R intervals without any exclusions using the interpolation method described in prior work below. Specifically, the excluded intervals in our results follow the error type 4 and 5 in the prior work, where M intervals in our results correspond to $N \neq M$ intervals in the ground truth. We interpolate them into $\max(M,N)$ intervals evenly. The absolute median and 90 percentile error raised from 28ms and 75ms to 32ms to 89ms respectively.

We report this in lines 538-544: “Excluding unmatched R-R intervals, however, might lead to more optimistic results since the mismatch is likely due to poor signal quality. To understand the effect of this exclusion, we included all the R-R intervals for the healthy cohort and compared the two data streams using the interpolation method in [58]. The excluded intervals above follow the error type 4 and 5 in [58], where M intervals in our results correspond to $N \neq M$ intervals in the ground truth. We interpolated them into $\max(M,N)$ intervals evenly. This increased the absolute median error from 28 ms to 32 ms and the 90th percentile error from 75 ms to 89 ms.”

[58] Gamelin, F-X., et al. "Validity of the polar S810 to measure RR interval." International journal of sports medicine 29.02 (2008): 134-138.

Reviewer 3:

The ability to monitor heart beats using smart speakers is very useful. It is very challenging since heart beats cause very tiny movements. There have been several previous works including those cited by the authors that can monitor breathing using smart speakers. But heart beats lead to even smaller movements, and it is surprising smart speakers can pick up the movements. The user studies across different people including the effect of obesity is interesting.

We thank the reviewer for their positive comments.

The authors could elaborate on the details for the beamforming design. Their description suggested they used DNN. What is the DNN structure? How much training data is required to train the DNN? How well the DNN generalizes?

We apologize for the confusion: we do not use a deep neural network (DNN). We also do not use any training data with ground truth information. As mentioned above, we are not performing training in the conventional sense where one provides ground truth data in the training phase. We do not use any ground truth data for identifying the weights of our beamformer. Our optimization is self-supervised, which means that the inference for one person does not require ground truth training data for the person or pretrained model on other people. The self-supervised model extracts the hidden information (i.e., the RR intervals) by optimizing an objective function with constraints using the test data from itself. The reason we use self-supervision is that different body shape, position and the surrounding environment make a supervised model difficult to generalize. Since we do not use ground truth data for learning the weights and since the weights are different for each position/person, cross-correlation is not appropriate for our algorithm. We are identifying the beamforming weights that maximize the signal strength of the heart rhythm motion by solving our optimization problem, without the need for any ground truth training data. The analogy is traditional (per-neural networks) beamforming algorithms that use an optimization function to maximize SNR, without any ground truth training data.

To address this confusion, we added lines 475-482: "Finally, our algorithm does not use supervised learning in that it does not need ground truth data. Our optimization is self-supervised, which means that the inference for one person does not require ground truth training data for the person or pre-trained model on other people. The self-supervised model extracts the hidden information (i.e., the R-R intervals) by optimizing the above objective function. The reason we use self-supervision is that different body shapes, positions and the surrounding environments make a supervised model difficult to generalize. Instead, we identify the beamforming weights that maximize the signal strength of the heart rhythm motion by solving our optimization problem, without the need for any ground truth training data."

The gradient ascent algorithm that is used for our optimization function in Eq. 1 is solved using standard Pytorch libraries as we describe in line 469-474.

Some heart patients have heart beats below 60 CPM. How does the current design handle this case?

Our beamformer is designed under the assumption that the heart beats generally fall into 60-150 CPM. It is not a hard threshold and the band-pass filter we use can still pass signals below 60 CPM. In our evaluation, our system can accurately measure the heart rate as well as RR-intervals for patients with 50-60 CPM average heart rate as shown in Figure 3A. It may however not detect any signal or amplify spurious noise in the 60-150 CPM range if the heart rate is much lower. We will mention this limitation in the discussion section. Thanks for the comment.

Specifically, we added the following to lines 263-266: "Our beamformer algorithm assumes that the average heart rate falls in the 60-150 beats per minute range. This is not a hard threshold and our band-pass filter can still detect cardiac signals below 60 beats per minute (Figure 3A). If the heart rate is much lower, it may however not detect any cardiac signal or may amplify spurious noise."

Clothes have significant impacts. Loose clothes even if thin can significantly reduce the accuracy. Do you see that?

Thank you for this comment. We did not restrict the clothing choices of the participants at the time of data collection.

With regards to the healthy participants, many of the participants did not wear tight fitting clothes. If they wore an extra layer of clothes (e.g., sweater /jacket) during the experiments, we requested them to take the 2nd layer off. With regards to hospitalized patients, they all wore hospital gowns which were a loose fitting single layer clothing.

To better quantify the impact of the looseness of the clothes, we did an extra experiment on two healthy participants to evaluate the difference in the accuracy when wearing a loose layer of clothes versus a tight layer of clothes. There was a slight change in accuracy with loose clothing: the median and 90 percentile absolute R-R interval errors decreased from 24 ms to 26ms and 84ms to 80ms respectively).

To address this comment, we added lines 270-276: "Performance results with the healthy cohort showed the system's reliability across diverse participant clothing, which included a single layer of shirts and tops that were not tightly fit; and many of the hospitalized patients wore loose gowns. While loose clothes can affect accuracy, the degradation was not drastic: the median and 90 percentile absolute R-R interval errors changed from 24 ms to 26 ms and 84 ms to 80 ms respectively for two participants who participated with both tight and loose clothes. However, multiple layers of clothing can limit the ability to extract heart motion since sound attenuates through thick fabric."

There is a closely related paper and it'd be good to compare with: Contactless Seismocardiography via Deep Learning Radars. Unsoo Ha, Salah Assana ; Fadel Adib. MobiCom 2020.

We thank the reviewer for their suggestions. We note that this work, while very interesting, is radar based and is demonstrated using healthy participants. Thank you for the suggestion; we have added this citation and discussed radar-based studies in the discussion section in lines 299-303: "Radar-based systems use radio signals with large bandwidth and use custom hardware that is not pervasive in smart speakers. Prior studies report a median error in R-R intervals of 8-44 ms for healthy participants (13,46-48) and 186 ms for cardiac patients with atrial fibrillation (10). Our sound-based system instead uses active sonar algorithms, hardware that is pervasive in smart speakers and is designed to achieve low errors for both regular and irregular rhythm."

We again thank all the reviewers for their time and their contributions in improving our manuscript.

REVIEWERS' COMMENTS:

Reviewer #1 (Remarks to the Author):

I am satisfied that the revised version of this manuscript answers my comments.

Reviewer #2 (Remarks to the Author):

The authors' answers are very satisfactory. I can recommend a publication and wish the authors much success in their further work.